# Light-induced charge generation in polymeric nanoparticles restores vision in advanced-stage retinitis pigmentosa rats

S. Francia [1,2,10], D. Shmal [1,3,10], S. Di Marco [1,2,10], G. Chiaravalli[4], J. F. Maya-Vetencourt [1,5], G. Mantero [1,3], C. Michetti [1,3], S. Cupini [1,3], G. Manfredi[4,6], M. L. DiFrancesco [1,2], A. Rocchi [1], S. Perotto [4], M. Attanasio[7], R. Sacco[8], S. Bisti[1], M. Mete[7], G. Pertile [7], G. Lanzani [4,9✉], E. Colombo [1,2,10] & F. Benfenati [1,2,10✉]

Retinal dystrophies such as *Retinitis pigmentosa* are among the most prevalent causes of inherited legal blindness, for which treatments are in demand. Retinal prostheses have been developed to stimulate the inner retinal network that, initially spared by degeneration, deteriorates in the late stages of the disease. We recently reported that conjugated polymer nanoparticles persistently rescue visual activities after a single subretinal injection in the Royal College of Surgeons rat model of *Retinitis pigmentosa*. Here we demonstrate that conjugated polymer nanoparticles can reinstate physiological signals at the cortical level and visually driven activities when microinjected in 10-months-old Royal College of Surgeons rats bearing fully light-insensitive retinas. The extent of visual restoration positively correlates with the nanoparticle density and hybrid contacts with second-order retinal neurons. The results establish the functional role of organic photovoltaic nanoparticles in restoring visual activities in fully degenerate retinas with intense inner retina rewiring, a stage of the disease in which patients are subjected to prosthetic interventions.

[1] Center for Synaptic Neuroscience and Technology, Istituto Italiano di Tecnologia, Genova, Italy. [2] IRCCS Ospedale Policlinico San Martino, Genova, Italy. [3] Department of Experimental Medicine, University of Genova, Genova, Italy. [4] Center for Nanoscience and Technology, Istituto Italiano di Tecnologia, Milano, Italy. [5] Department of Biology, University of Pisa, Pisa, Italy. [6] Novavido s.r.l., Bologna, Italy. [7] Department of Ophthalmology, IRCCS Sacrocuore Don Calabria Hospital, Negrar, Verona, Italy. [8] Department of Mathematics, Politecnico di Milano, Milano, Italy. [9] Department of Physics, Politecnico di Milano, Milan, Italy. [10] These authors contributed equally: S. Francia, D. Shmal, S. Di Marco, E. Colombo, F. Benfenati. ✉email: guglielmo.lanzani@iit.it; fabio.benfenati@iit.it

Retinal dystrophies, such as *Retinitis pigmentosa* (RP) are the most common inherited progressive cause of blindness in developed countries affecting some 5.5 million patients[1,2] and for which effective therapeutic strategies are in demand. Gene replacement/supplementation therapy, optogenetics or cell therapies have made considerable progress. A gene therapy is commercially available for RPE65-linked retinal dystrophies, while clinical trials for a restricted group of other RP genes are ongoing[3]. However, it is unlikely that gene replacement therapy could cover the entire genetic landscape of RP that likely involves more than 200 genes[4–6]. Optogenetics is an alternative mutation-independent gene therapy whose efficacy has been widely tested in preclinical models and recently reported in one RP patient within an ongoing clinical trial[7,8]. However, the inherent low sensitivity of the heterologous opsins requires an external camera coupled to light intensification goggles, resulting in some limitations in performance[8,9]. Other cell-based approaches are still in preclinical experimentation or early phases of human testing[10–12]. We recently reported that the subretinal injection of conjugated polymer nanoparticles (P3HT-NPs) rescues visual functions in the Royal College of Surgeons (RCS) model of RP[13]. This rat strain bears a mutation in the *Mertk* gene[14], the very same gene whose mutation causes a *cone-rod dystrophy* in humans characterized by fast progression due to the early and simultaneous involvement of both photoreceptor types[15–18]. P3HT-NPs were administered to 3-months-old RCS rats, an age at which the rod/cone degeneration was almost complete, and the visual rescue was stable for at least 8 months after a single subretinal injection[13]. Although, at the time of the injection, sparse photoreceptors were still present in the presence of an apparently intact inner retina, no increased photoreceptor survival or slowdown of their degeneration was observed, supporting a direct action of P3HT-NPs onto second-order retinal neurons.

While gene and cell therapies are most effective in the early stages of the disease trying to counteract the ongoing degeneration, retinal prostheses have been developed to electrically stimulate the inner retinal neurons that are not primarily involved in the degenerative process and are generally applied in advanced stages of the disease[19–21]. A severe limitation to the success of retinal prosthetics is the structural and functional rewiring of the inner retina. Inner retinal circuits, spared by degeneration, respond to the chronic deafferentation from photoreceptor inputs with profound rearrangements and tend to deteriorate in the late stages of the disease, i.e., in the temporal window in which the prosthetic therapy is generally attempted[22,23]. It has been reported that, in a variety of animal models of RP, including the RCS rats, bipolar and horizontal cells, missing the synaptic inputs from photoreceptors, undergo dendritic sprouting. Moreover, in the inner retina, the synaptic contacts between bipolar, amacrine and ganglion cells are dramatically changed, leading to abnormal processing and transfer of visual information[24–33]. Numerous reports confirmed that these remodeling processes of the inner retinal circuitry also occur in patients with RP. Although the cellular mechanisms of the human retina rewiring are not entirely understood, an extensive involvement of the inner retina (bipolar cells, amacrine) and of glial and Müller cells has been described since the beginning of retinal degeneration, consisting of cellular reprogramming and early alterations of the retinal signaling proteins[34–39].

Although our previous experiments demonstrated a role of P3HT-NPs in promoting a sustained visual restoration in the rat[13], the possibility remains open that reactivating the vision process at an early stage of the disease could somehow interfere with the degeneration itself and be highly facilitated by the presence of a still functional inner retina. Operating a fully degenerated and aged retina, which is indeed the realistic case in the human patient, brings additional challenges. Here, we demonstrate that subretinal P3HT-NPs are capable to restore vision under conditions of long-standing degeneration that had led to complete photoreceptor death and complex remodeling/rewiring of the inner retina, with total absence of visual responses. Injected in 10 months old RCS rats, P3HT-NPs restored physiological pupillary light reflex (PLR), visually evoked cortical potentials (VEPs) and visually driven behavior to the levels of healthy, age-matched congenic animals, despite the total absence of photoreceptors and the presence of a fully remodeled inner retina. In addition, P3HT-NPs were also able to partially restore visual acuity and reactivate the formation of light-driven implicit memory traces integrated at multiple cortical levels. This outcome not only fully validates the visual restoration activity of P3HT-NPs, but has also an important prognostic meaning on the potential success of a prosthetic intervention at a RP advanced stage, which may also apply to prosthetic devices other than P3HT-NPs.

## Results

**The inner retina of RCS rats becomes profoundly remodeled with ageing.** Subretinal prostheses have the advantage of using the inner retinal circuits to process elementary light-induced signals before their delivery to the brain. Yet it is well known that the inner retina, although spared by neurodegeneration, undergoes complex rearrangements in the absence of photoreceptor input. Thus, we first characterized this remodeling process over age by studying retina morphology and the expression of specific biomarkers of the retinal neuronal populations in young (2 month-old) and old (13/15-month-old) dystrophic RCS rats, comparing them with healthy age-matched congenic RCS-rdy (rdy) rats. Optical coherence tomography (OCT) scans showed that the retina thickness is markedly decreased in RCS rats with respect to rdy controls due to photoreceptor degeneration already in young animals, as previously reported[34] and further progresses toward total external retina atrophy in old animals (Fig. 1a; Supplementary Fig. 1). The analysis of the expression of specific markers of the retinal neuronal populations by quantitative reverse transcription PCR (qRT-PCR) revealed that the messenger RNAs (mRNAs) of photoreceptor opsins (*Rho*, *Opn1sw* and *Opn1mw*) are markedly downregulated in young RCS with respect to healthy age-matched congenic rdy rats and become undetectable in old RCS animals. On the contrary, the mRNA levels of protein kinase Cα (*Prkca*, specific marker of bipolar cells) and calbindin1 (*Calb1*, specific marker of horizontal cells) remain relatively unaffected in young and old RCS and rdy controls (Fig. 1b). Immunohistochemical analysis confirmed these findings in greater detail. The population of rhodopsin-labeled rods was already absent in young RCS (both as cell body counts and outer segment length), except for occasional degenerating rod cell bodies that totally disappeared in old RCS (Fig. 1c). Cone arrestin-labeled cones displayed a similar pattern, with few remaining cell bodies in the outer nuclear layer (ONL) that completely lost the external segment, and a nearly total disappearance in old RCS (Fig. 1d). On the contrary, the number of PKCα-labeled rod bipolar cells (rBPCs) and Calbindin1-labeled horizontal cells (HzCs) in the inner nuclear layer (INL) was fully preserved across all experimental groups, notwithstanding the dramatic retinal dystrophy of RCS rats. Despite the substantial preservation of its components, the cytoarchitecture of the inner retina was markedly altered: rBPCs lost their radial orientation in RCS rats, as shown by the increase of their axon angle that was already significant in young animals and further progressed in aged animals, and HzCs become progressively closer to the retinal pigment epithelium (RPE) (Fig. 1e,f). Together with the nuclear

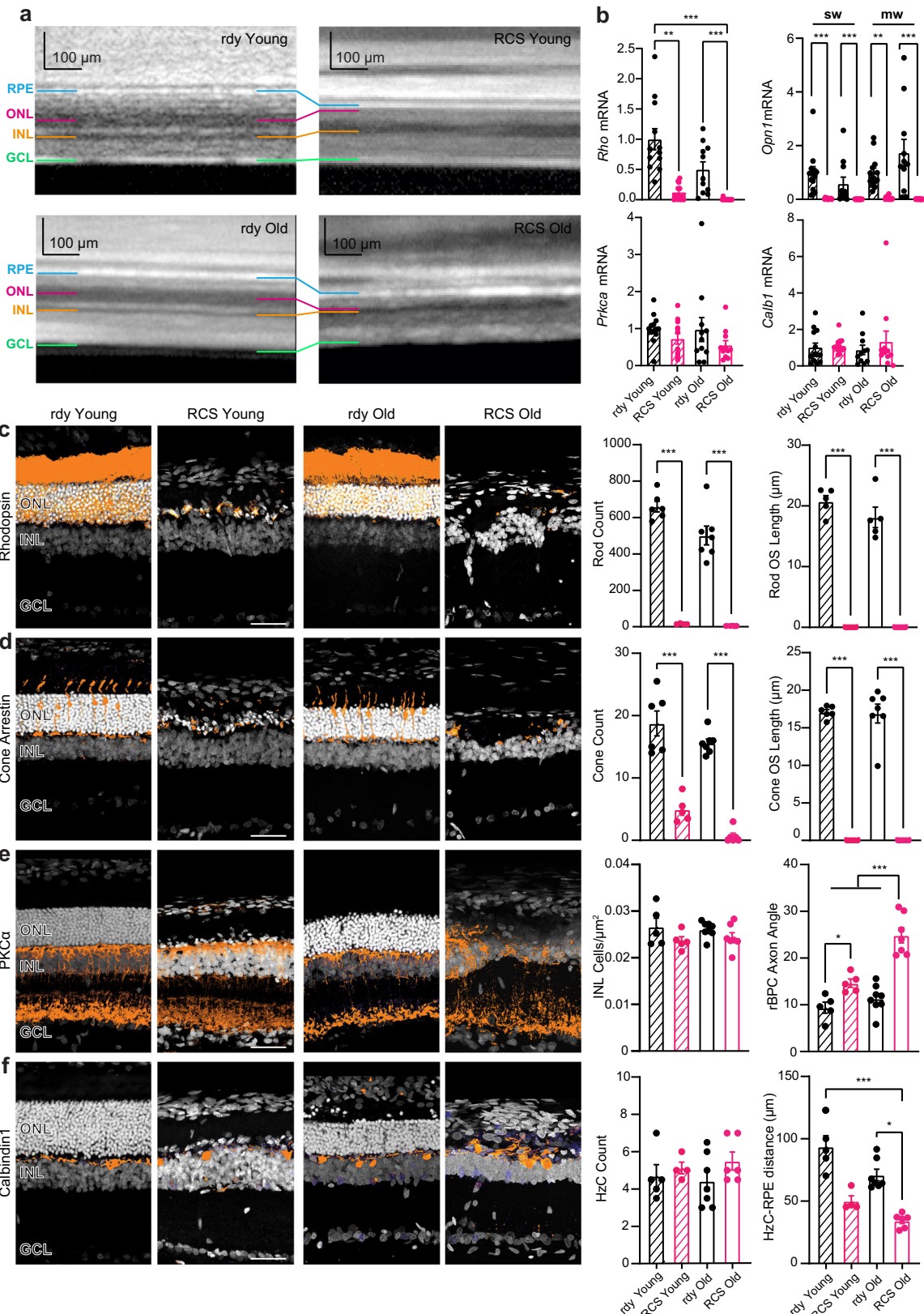

staining with bisbenzimide (white; Fig. 1c–f), the immunohistochemical data testify to the intense rearrangement of the inner retina that, from an initial integrity in early stages of photoreceptor degeneration, progresses over time towards a fully disordered architecture.

**P3HT-NPs distribute evenly in the subretinal space of both young and old RCS rats.** P3HT-NPs were obtained by the reprecipitation method in the absence of surfactant, as previously reported[40,41] and checked by scanning electron microscopy (SEM) and dynamic light scattering (DLS). SEM revealed a

**Fig. 1 Fast photoreceptor degeneration in *Mertk* mutant RCS rats triggers a progressive rewiring of the inner retina over aging. a** Representative optical coherence tomography (OCT) images of young (2-month-old) and aged (13/15-month-old) dystrophic RCS retinas as compared to age-matched healthy congenic rdy rats. Retinal layers are lined in different colors: retinal pigmented epithelium (RPE) in blue, outer nuclear layer (ONL) in red, inner nuclear layer (INL) in orange, ganglion cell layer (GCL) in green. The ONL is completely missing in dystrophic RCS rats at both ages. Replicates: $n = 6,5,7,6$ for rdy Young RCS Young, rdy Old, RCS Old respectively. Scale bars, 100 μm. **b** mRNA levels of Rhodopsin (*Rho*), Opsin-1 short wave-sensitive (*Opn1sw*), Opsin-1 medium wave-sensitive (*Opn1mw*), Protein kinase C-alpha (*Prkca*) and Calbindin1 (*Calb1*) were quantified by RT-qPCR in retinal sections dissected from dystrophic RCS and non-dystrophic rdy rats at 2 (*Young*) and 13/15 months (*Old*) of age. *Gapdh* and *Pgk1* were used as control housekeeping genes. One-way Kruskal–Wallis ANOVA/Dunn's test. Sample sizes: *Rho* 12,12,11,11; *Opn1sw* 13,12,11,11; *Opn1mw* 13,12,11,11; *Prkca* 13,12,11,11; *Calb1* 13,12,11,11; for rdy Young, RCS Young, rdy Old and RCS Old, respectively. **c–f** *Left:* Transversal sections of retinas dissected from dystrophic (RCS) and non-dystrophic (rdy) rats at 2 months (Young) and 13/15 months (Old) of age. Sections were immunolabelled with antibodies to rhodopsin for rod photoreceptors (**c**), cone arrestin for cone photoreceptors (**d**), protein kinase C-alpha (PKCα) for rod bipolar cells (rBPCs) (**e**), and calbindin1 (Cb1) for horizontal cells (HzCs; **f**). Scale bar, 50 μm. *Right:* rod and cone numbers per field and outer segment (OS) lengths, deviation of rBPC axons from the physiological right angle to the GCL and rBPC density in the INL, distance of HzC bodies from the RPE and HzC counts per field. One-way ANOVA/Holm-Šídák's or Kruskal–Wallis/Dunn's tests. Sample sizes: rod counts 6,4,7,6; rod OS length 5,4,5,6; cone counts 6,5,7,6; cone OS length: 6,5,7,6; rBPC axon angle and INL density 5,5,8,7; HzC-RPE distance and counts 5,4,7,6; for rdy-RCS (Young), rdy-RCS (Old), respectively. Graphs show means ± sem with superimposed individual points. *$p < 0.05$, **$p < 0.01$, ***$p < 0.001$. For exact *p*-values and source data, see Source data file.

homogeneous population of NPs, with DLS spectra showing an average diameter of 182 nm and ξ potential with a peak at −21 mV (Supplementary Fig. 2). First, we studied the retinal distribution by injecting P3HT-NPs into the subretinal space of RCS rats at 2 and 10 months of age. After the injection, the retinas of all experimental groups (young/old RCS) were monitored by OCT and displayed a very efficient retina reattachment with substantial integrity of the retinal layers (Supplementary Fig. 3). The distribution of subretinally injected fluorescent $SiO_2$-NPs (fluoSiO$_2$-NPs) of the same size of P3HT-NPs was also assessed, given that inert $SiO_2$-NPs will be used in sham-operated animals for the subsequent functional studies.

One month after injection, the distribution and dispersion of NPs were studied by super-resolution confocal microscopy in excised whole-mount retinas stained with bisbenzimide (cell nuclei), taking advantage of the intrinsic fluorescence of the NPs. When microinjected under the pigmented epithelium of dystrophic retinas, both P3HT-NPs and fluoSiO$_2$-NPs widely distributed in the subretinal space (Fig. 2a). NPs remained confined to the outer retina with no tendency to radially diffuse to the inner retina layers (Fig. 2b). Morphometric analysis of z-stack confocal images using high-dynamic-range fluorescence microscopy (Leica SP8/HyD with Lightning deconvolution, see Methods) in whole-mount and retinal sections showed that both P3HT- and fluoSiO$_2$-NPs reached an appreciable density in most retinal regions, although their density was maximal around the site of injection (Fig. 2a). The same analysis revealed that both P3HT- and fluoSiO$_2$-NPs were similarly organized in small microaggregates with skewed distribution and maximum frequency (40 to 60% of the total P3HT-NPs) at a diameter < 0.5 μm (Fig. 2c, left). High-content morphometric analyses performed on retinal sections calculated that the extent of retina coverage by both P3HT- and fluoSiO$_2$-NPs was over 60% (Fig. 2c), in the same range as the previous reports in younger animals[13], with a nearest neighbor distance of about 5 μm, in the same range as the size of macular cones (Fig. 2c, right). These values did not significantly differ between P3HT- and fluoSiO$_2$-NPs, confirming the goodness of SiO$_2$-NPs as control NPs for sham-injected animals. We also checked whether the distribution of P3HT-NPs was different at distinct stages of degeneration by studying young (3-month-old) *versus* aged (13/15-month-old) retinas (Supplementary Fig. 4a). Results show that P3HT-NPs were similarly distributed with ~95% of the clusters characterized by diameters < 2 μm, indicating a high degree of dispersion in both young and old dystrophic retinas. Only <1% of the particles gave rise to microaggregates with diameter >5 μm (Supplementary Fig. 4b).

**Subretinally injected P3HT-NPs contact bipolar and horizontal cells and undergo a light-induced build-up of charge polarization**. We verified whether P3HT-NPs, subretinally injected in the massively rewired retinas of old RCS rats, could contact second-order retinal neurons, such as rBPCs and HzCs, that are responsible for inner retinal processing and retinal ganglion cell (RGC) activation. Using high-dynamic range, hybrid detector-based fluorescence microscopy on retinal sections stained with the rBPC- and HzC-specific markers PKCα and Calbindin1, respectively (Fig. 3a–c), we could indeed show significant overlaps between the P3HT intrinsic fluorescence and the immunoreactive areas of rBPCs and HzCs, as evaluated by the Manders' M1 and M2 coefficients (Fig. 3d). Moreover, using 3D reconstruction of retinal layers, we detected high relative densities of P3HT-NPs in the INL layer, as well as number of direct contact points of P3HT-NPs with cell bodies and processes of rBPCs and HzCs (Fig. 3e), yielding basic connectivity between P3HT-NPs and second-order retinal neurons.

Photostimulation by P3HT-NPs was proposed to consist in the photo-induced electrical polarization and subsequent capacitive coupling of the NPs in tight "*gigaseal*" contact with the neuronal membrane[13]. Here we explored the mechanism of charge separation in P3HT-NPs and their photostimulation action. Long-lived separated charge pairs are formed in the crystalline domains of the NP, as expected for regio-regular P3HT[42]. The essential physics is well-captured by the 1D-numerical solution of the drift-diffusion model in the NP, coupled to Poisson-Nernst-Planck equations for the ions in the cleft and extracellular medium. The initial Lambert-Beer profile within the NP drives carrier diffusion, leading to an excess of negative charge at the illuminated surface and an excess of positive charge at the opposite side because of the well-known asymmetry in carrier mobility: holes spread out homogeneously in the film, while electrons (negative polarons) are essentially immobile. The separation in space of the photo-charge carriers gives rise to the electrical polarization of the NP (see Supplementary Text and Supplementary Figs. 5–10). By modeling the NP-aqueous environment interface assuming that (i) oxygen reduction takes place at the negatively charged surface[43,44] and (ii) the cleft between the NP and the neuronal membrane[13] consists of a highly resistive protein medium (Fig. 3f), the reduced oxygen ion distribution is predicted to give rise to a sizable electrical potential. The model does not depict the complex interface with the neuron, yet it suggests that the spatial range of the electrical potential generated at NP surface is in principle capable of capacitively depolarize the membrane of second-order retinal neurons.

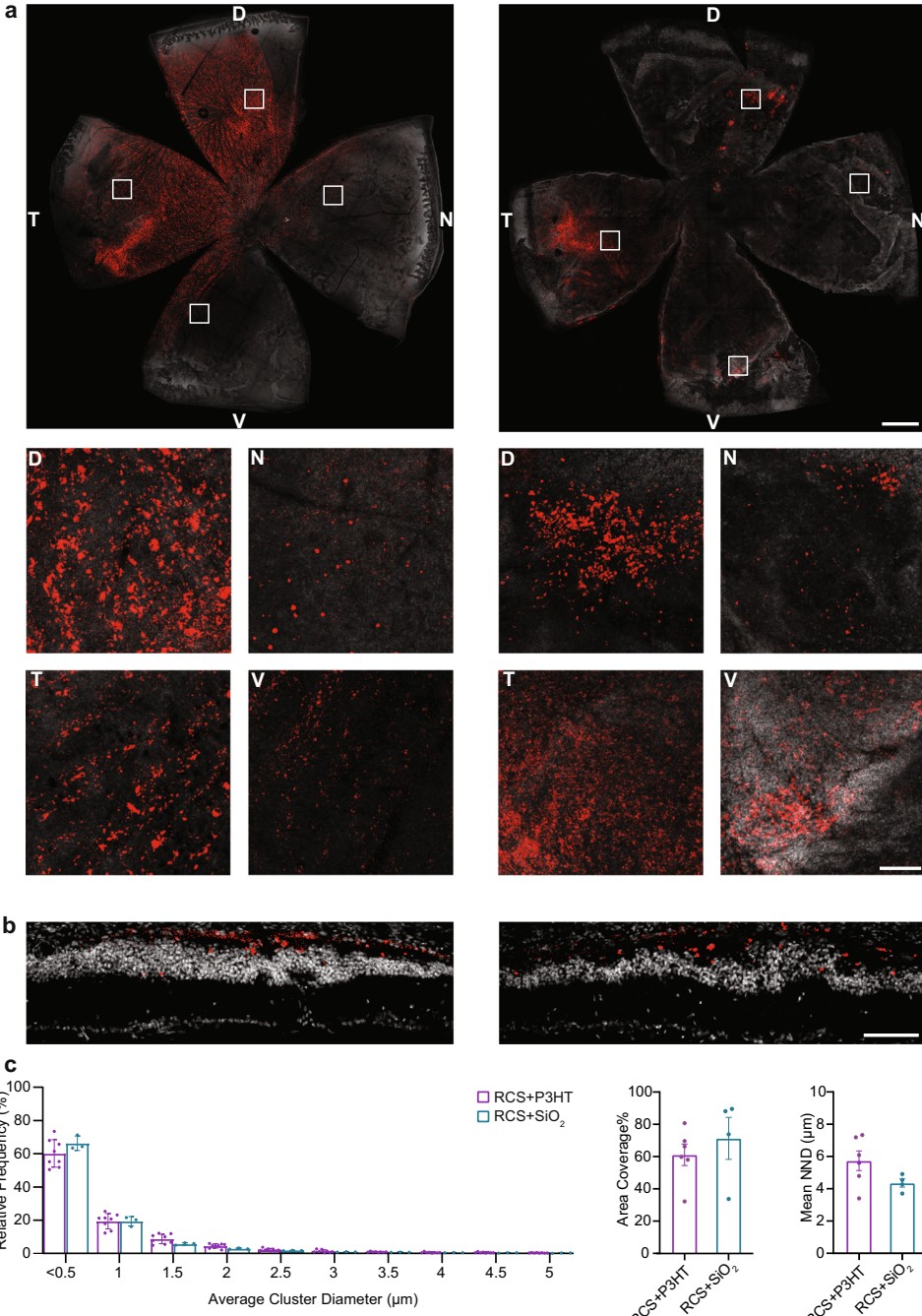

**Fig. 2 Distribution of fluorescent P3HT- and SiO$_2$-NPs after subretinal injection in old dystrophic RCS rats. a** Whole-mount retinas from 13/15-month-old RCS rats were analyzed by confocal z-stack scans in which the localization of P3HT-NPs (*left*) and SiO$_2$-NPs (*right*) (intrinsic fluorescence in red) is clearly visible with respect to retinal cells (nuclear staining with bisbenzimide in white). Each panel represents the z-max with local magnifications of dorsal (D), nasal (N), ventral (V) and temporal (T) regions of the retina, highlighting the degree of dispersion of the P3HT- and fluorescent SiO$_2$-NPs. Scale bar, 1 mm, high magnification areas 0.5 × 0.5 mm. Inset scale bar, 100 μm. **b** Transversal retinal sections from dystrophic RCS rats injected with either fluorescent P3HT-NPs or fluorescent SiO$_2$-NPs at 30 DPI. NP fluorescence (red) was merged with bisbenzimide nuclear staining (white). The images emphasize the stable subretinal location of injected NPs, with no tendency to permeate the internal retinal layers. Scale bar,100 μm. **c** Histogram showing the frequency distribution of average cluster diameters (*left*), the extent of retina coverage (*middle*) and the nearest neighbor distance (NND; *right*) for P3HT-NPs and fluorescent SiO$_2$-NPs. The estimated diameter of each NP cluster was calculated as the average of the three dimensions of each isolated fluorescence volume's bounding box. About 40–60% of the volumes have diameters below 500 nm in all cases, with an average of almost monodisperse NPs (>50%) regardless of age. Data are shown as means ± sem with individual experimental points. Sample size: $n = 3$ (RCS + SiO$_2$) and 8 (RCS + P3HT) for cluster volumes, $n = 4$ (RCS + SiO$_2$) and 6 (RCS + P3HT). Kolmogorov-Smirnov and Mann–Whitney $U$ tests. For exact $p$-values and source data, see Source data file.

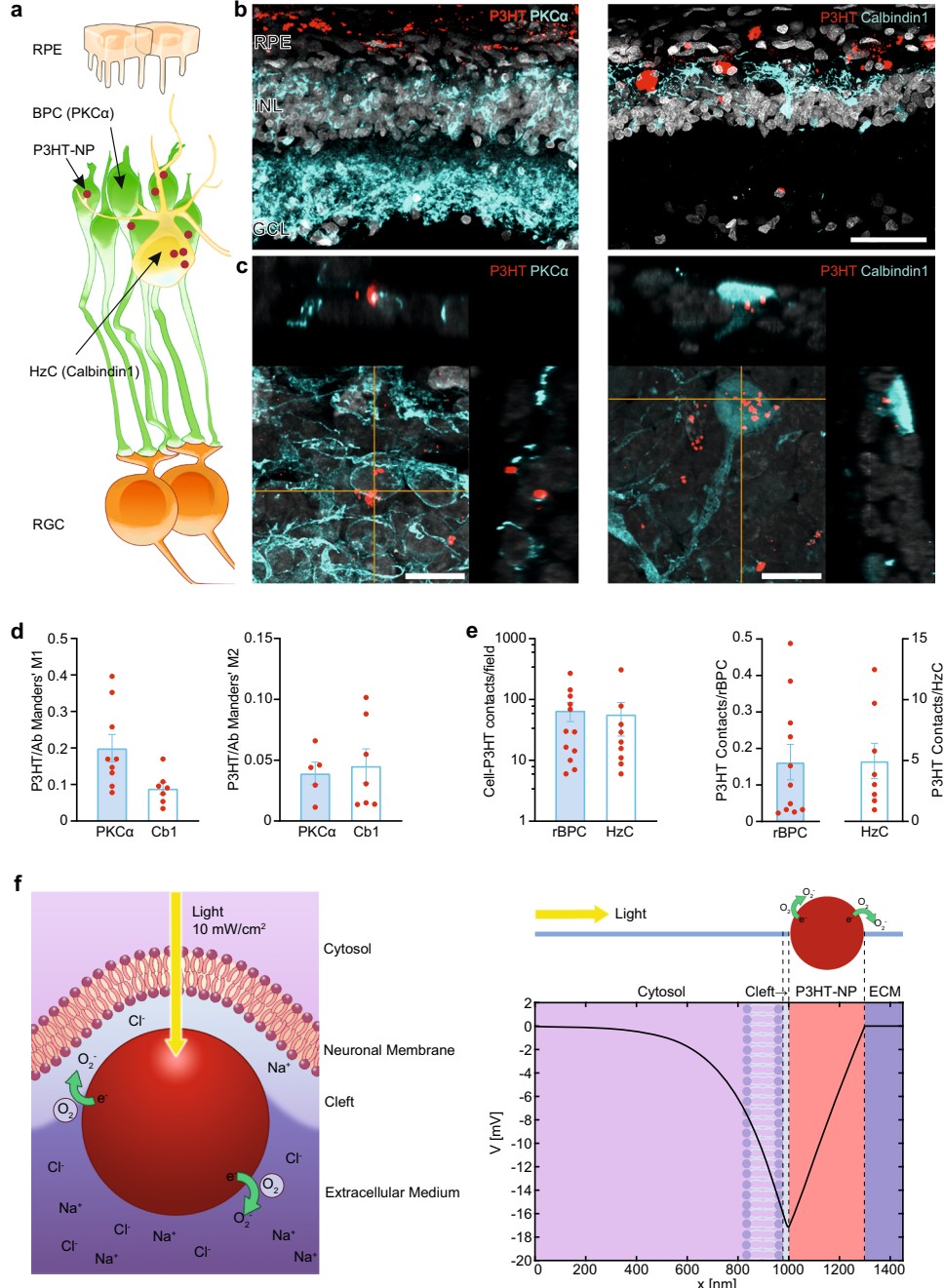

**Fig. 3 P3HT-NPs form hybrid contacts with bipolar and horizontal cells. a** Schematic representation of second-order retinal neurons. rBPCs and HzCs were identified by immunofluorescence targeting the specific markers PKCα and Calbindin1, respectively. **b** Representative transverse sections of 13/15-month-old P3HT-NP injected RCS animals stained for rBPCs or HzCs (blue). The P3HT-NPs intrinsic fluorescence is in red. Scale bar, 50 μm. **c** Representative z-max projections with x- and y-orthoslices of super-resolution confocal images showing close contacts of P3HT-NPs with rBPCs and HzCs. Scale bar, 10 μm. **d** Bar plots (means ± sem with individual data points) showing Manders' M1 (expressing the proportion of the total P3HT fluorescent area colocalizing with rBPC/HzC immunoreactive area) and M2 (expressing the proportion of the total rBPC/HzC immunoreactive area colocalizing with the P3HT-positive area) coefficients for both markers. Values were thresholded at 0.01. M1: $n = 9$ (PKCα) and 7 (Cb1); M2: $n = 5$ (PKCα) and 7 (Cb1). **e** Average number of NP clusters contacting rBPCs and HzCs per field and per average rBPC and HzC count. Data are shown as means ± sem with individual experimental points. **b–e** Sample size: 12 (rBPC), 9 (HzC) per field; $n = 11$ (rBPC), 8 (HzC) per cell. rBPC, rod bipolar cell; HzC, horizontal cell. **f** Left: Schematics of a P3HT-NP microenvironment comprising the nanoparticle (*red*), the cleft (light blue) and the extracellular environment (*purple*). Due to the extreme*ly narrow* cleft (<20 nm), the bulk of the extracellular electrolyte is excluded from the NP-neuronal membrane junction. Right: Electric potential across the 1D computational domain comprising the cleft, the NP and the extracellular medium. Light naturally entering the eye hits the NP from the cleft side in which the NP contacts second-order retinal neurons. Source data are provided for panel **d**, **e** as a Source data file.

**P3HT-NPs do not alter the inner retina or elicit inflammatory responses in aged RCS rats**. We next addressed the central question of whether P3HT-NPs were still effective in rescuing visual performance in aged dystrophic RCS rats characterized by the total absence of photoreceptors and complex inner retina remodeling. The experimental strategy enrolled four groups of animals aged 10 months at the time of injection, namely untreated rdy controls, untreated RCS rats, sham-injected RCS rats (injection of inert size-matched SiO$_2$-NPs) and P3HT-NP injected RCS rats. The timeline of the experiments, including the physiological analyses of visual rescue, is depicted in Supplementary Fig. 11.

Surgery has been reported to have trophic effects[45–47] by inducing the release of growth factors that may attenuate photoreceptor degeneration and affect retinal circuits. Although in the advanced disease-stage animals used in this study, no traces of residual photoreceptors were present, age-matched untreated and sham-injected RCS rats, subjected to the same surgery and injection of inert SiO$_2$-NPs, were included to have full control over potential off-target effects of surgery or of the presence of NPs as foreign bodies. At 90–150 days after injection, we comparatively analyzed the mRNA levels for photoreceptors (*Rho*, *Opn1sw* and *Opn1mw*), BPCs (*Prkca*) and HzCs (*Calb1*) by RT-qPCR (Supplementary Fig. 12). Photoreceptor markers were undetectable in all aged RCS rats, while the inner retinal markers were not affected by the microinjection of either P3HT-NPs or SiO$_2$-NPs with respect to the untreated RCS group. We also verified whether the microinjection of NPs did not have any effect on inner retina remodeling present in aged dystrophic retinas. The extent of RCS retina rewiring, evaluated by rBPC axon orientation and HzC-RPE distance, was not affected by either surgery or the presence of NPs. All RCS groups, irrespective of whether they were untreated or injected with either NPs, had significantly higher parameters of retinal disorder than age-matched rdy controls (Supplementary Fig. 13). Extensive subretinal lavage was previously shown to improve visual functions in very young (38-day-old) dystrophic animals by removal of the accumulated cell debris[44]. Although our surgical procedure for the subretinal injection of NPs did not involve subretinal lavage, we performed histological staining of the retinas from RCS rats which were untreated or injected with P3HT-NPs or SiO$_2$-NPs. In untreated RCS rats, the subretinal space, occupied by debris in early stages of degeneration, was markedly reduced in the advanced stage (Supplementary Fig. 14a; see also the reduction of HzC-RPE distance in Fig. 1). Under these conditions, no accumulation of cell debris was present in the outer retina in all the experimental groups irrespective of injection surgery (Supplementary Fig. 14b).

P3HT is known to be highly biocompatible and tissue-friendly[13,48,49]. However, we addressed the possibility that the injection of NPs has proinflammatory effects by analyzing GFAP and Iba-1 immunoreactivities as markers of astrocyte/Muller cell gliosis and microgliosis, respectively. With respect to rdy controls, dystrophic retinas were characterized by an intense astroglial reaction, which was not, however, significantly affected by the injection of either SiO$_2$- or P3HT-NPs. Astrogliosis was not significantly correlated with the P3HT-NP density, although the injection of NPs showed a trend for a decrease in GFAP immunoreactivity (Supplementary Fig. 15). Dystrophic retinas were characterized by an intense microglial reaction with respect to rdy controls (Supplementary Fig. 16a). The degree of microglia activation was evaluated in terms of number and extension of processes by Sholl analysis and cell body area. The analysis showed that microglial activation was similar across the experimental groups of dystrophic rats, with no signs of harmful reactive microglia in dystrophic retinas irrespective of the

treatments (Supplementary Fig. 16b–d). As in the case of astrogliosis, the microglial reaction of dystrophic rats was not significantly affected by the injection of either SiO$_2$- or P3HT-NPs and no significant correlation with the P3HT-NP density was observed (Supplementary Fig. 16e–g).

**P3HT-NPs restore light-triggered subcortical reflexes in aged RCS rats**. We first analyzed pupillary constriction as a proxy of light-sensitivity restoration. The PLR is mediated by retinal projections to the pretectal region of the brainstem and thereby is integrated at the subcortical level. It is triggered by the complementary action of intrinsically photosensitive RGCs expressing melanopsin (ipRGCs), sensitive to slow changes of intense blue light, and photoreceptors, driving the reflex at various wavelengths for lower luminances and fast stimuli[50].

To distinguish between the two components and study the complex dynamics of the PLR, we measured: (i) the latency of the pupillary response (PLR latency), (ii) the extent of constriction (PLR constriction), (iii) the constriction velocity, (iv) the extent of post-stimulus pupillary dilation (PIPR dilation) and (v) the dilation velocity in response to various intensities of green light (540 nm) stimuli (Fig. 4a). While PLR is driven by either photoreceptors or P3HT-NP input and by the response of intrinsically photosensitive melanopsin-positive RGCs, PIPR is a marker of the activity of ipRGCs and is used to infer their contribution to PLR (see Methods). At 20 lux stimulus illuminance, untreated dystrophic RCS rats were characterized by a longer PLR latency, a much smaller and slower constriction accompanied by an increased PIPR compared to the respective parameters recorded in rdy controls. RCS rats injected with P3HT-NPs fully recovered the reflex parameters to levels indistinguishable from those of healthy rdy rats, while RCS rats sham-implanted with inert SiO$_2$-NPs did not show any improvement with respect to untreated RCS rats (Fig. 4b–f; Supplementary Video 1). Similar results were obtained by varying the stimulus illuminance in the range of 5–50 lux (Supplementary Fig. 17).

**P3HT-NPs reinstate visually evoked potentials in the primary visual cortex of aged RCS rats**. To assess the restoration of light perception at the cortical level, we recorded VEPs in the binocular portion of V1 (OC1b) cortex of aged rdy controls and dystrophic (RCS) rats either untreated, microinjected with P3HT-NPs, or sham-injected with SiO$_2$-NPs (Fig. 5a, b)[13,49]. Compared to young animals, aged rdy controls exhibited attenuated VEP responses (20–30 µV) to light flash stimuli, consistent with a natural decrease in VEP amplitude occurring in albino rats with aging due to chronic light damage to the retina at the standard luminance used in animal facilities. On the contrary, age-matched dystrophic RCS had a total absence of cortical responses to the light stimuli, being the field potential oscillations largely within the 2xSD noise band. Notably, in RCS rats implanted with P3HT-NPs, the cortical responses to light stimuli revealed a restoration of both VEP amplitude and latency to the levels of age-matched rdy controls. On the other hand, sham-injected dystrophic animals did not show any rescue of cortical responses to the light stimuli that remained identical to untreated RCS rats (Fig. 5c). Importantly, at this stage of advanced and severe retina degeneration, with rewiring and regression of the inner retinal circuits, P3HT-NP injected RCS rats also recovered the VEP latency to the levels of healthy rdy controls (Fig. 5d). This directly implicates P3HT-NPs as the source of the electrical signals in the RGC axons that are sent to the brain from the implanted, fully degenerate retinas.

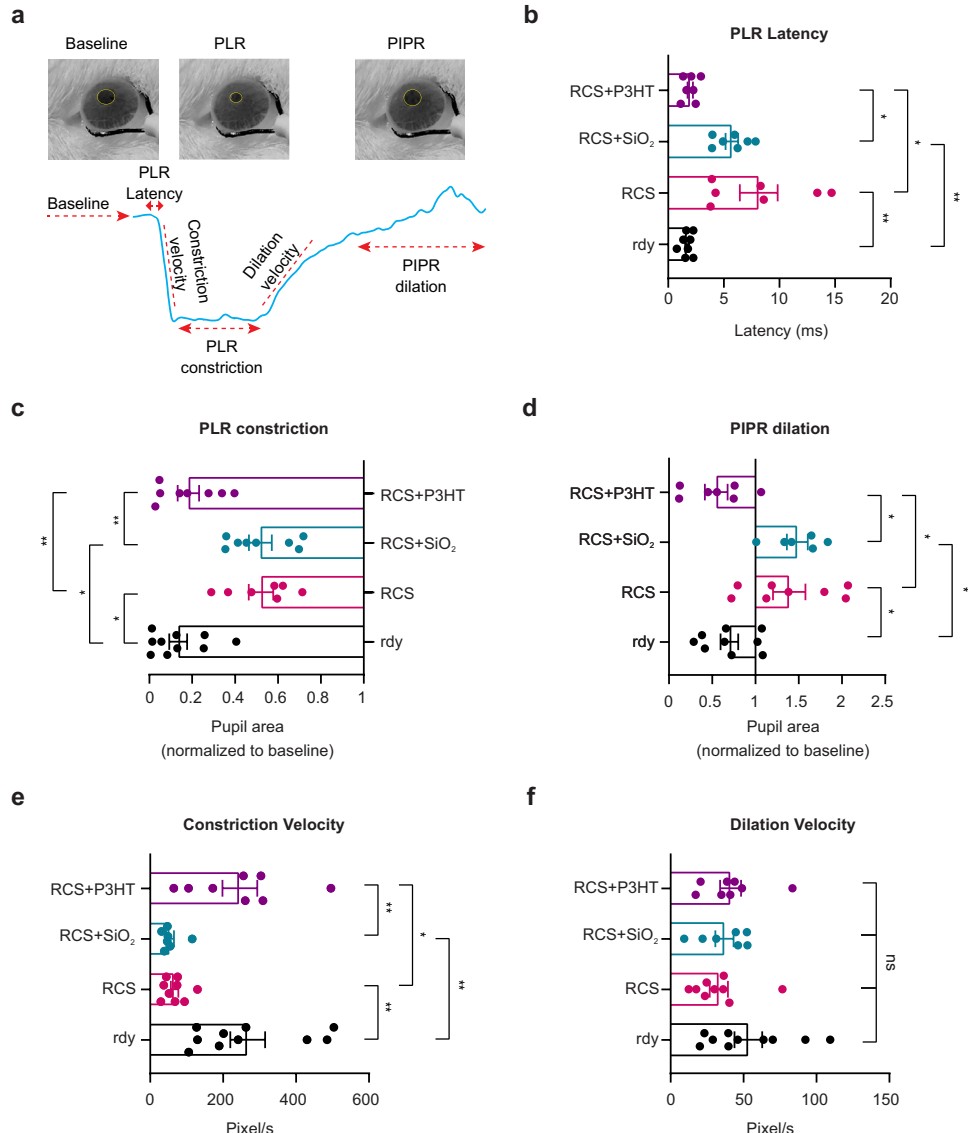

**Fig. 4 The pupillary light reflex is fully rescued by P3HT-NPs in aged RCS retinas. a** *Top:* The Pupillary Light Reflex (PLR) and Post-Illumination Pupillary Response (PIPR) were video recorded under infrared illumination in response to a prolonged stimulus (20 sec) of green light (530 nm) of 20 lux. Representative images of the pupil's maximal constriction are shown. *Bottom:* From the recorded video, we extrapolated: (i) the area of the pupil at baseline (before the light stimulus); (ii) the latency of the pupil constriction from the switch of the light to the start of pupillary constriction; (iii) the constriction velocity computed as the minimum first derivative value calculated during the constriction; (iv) the PLR constriction, as the average area of the pupil during the constriction normalized to the baseline area; (v) the dilation velocity, as the maximum value of the first derivative computed during the dilation phase; and (vi) the PIPR dilation, as the average area of the pupil during relaxation normalized to baseline. **b–f** Aged-matched (13/15-month-old) rdy, RCS, RCS + SiO$_2$ and RCS + P3HT were analyzed for PLR latency (ms, **b**), PLR extent of constriction (**c**), PIPR dilation (**d**), constriction velocity (**e**), and dilation velocity (**f**). RCS + P3HT animals performed like healthy rdy controls, with shorter constriction latency, a more intense pupillary constriction and a smaller relaxation than untreated RCS or sham-injected RCS rats. Kruskal–Wallis/Dunn's tests. Sample size: PLR Latency (7,8,7,7), PLR constriction (10,7,8,8), PIPR dilation (9,8,7,7), Constriction velocity (10,9,7,8), Dilation velocity (10,9,7,8) for rdy, RCS, RCS + SiO$_2$ and RCS + P3HT, respectively. Data are the means ± sem with individual experimental points. *$p < 0.05$, **$p < 0.01$. For exact *p*-values and source data, see Source data file.

**Light-driven behavioral responses are restored in aged dystrophic RCS rats injected with P3HT-NPs.** We next tested whether the rescue of cortical responses to light was reflected by the light-driven behavioral responses that rely on the innate aversion of rodents to illuminated areas and related anxiety[13,49]. Using dim light (5-lux) as a stimulus to challenge light sensitivity, the spontaneous behavior of the four experimental groups was evaluated by computing the latency of escape from the illuminated area to darkness and the percentage of time spent in the dark over the entire duration of the test (Fig. 6a). When the same RCS rats were measured for escape latency before and after the

subretinal injection of either P3HT-NPs or SiO$_2$-NPs, the previously absent light-escape response was rescued to the levels of healthy rdy controls only in RCS rats implanted with P3HT-NPs, but not with SiO$_2$-NPs (Fig. 6b, c). Strikingly, the overall analysis of the light-escape latency and preference for the dark compartment (Fig. 6d, e), revealed that aged P3HT-NP-injected dystrophic RCS animals significantly recovered their light-driven behavior to the levels of non-injected rdy controls, while the behavioral performances SiO$_2$-injected RCS rats were indistinguishable from those of non-injected RCS rats (Supplementary Video 2).

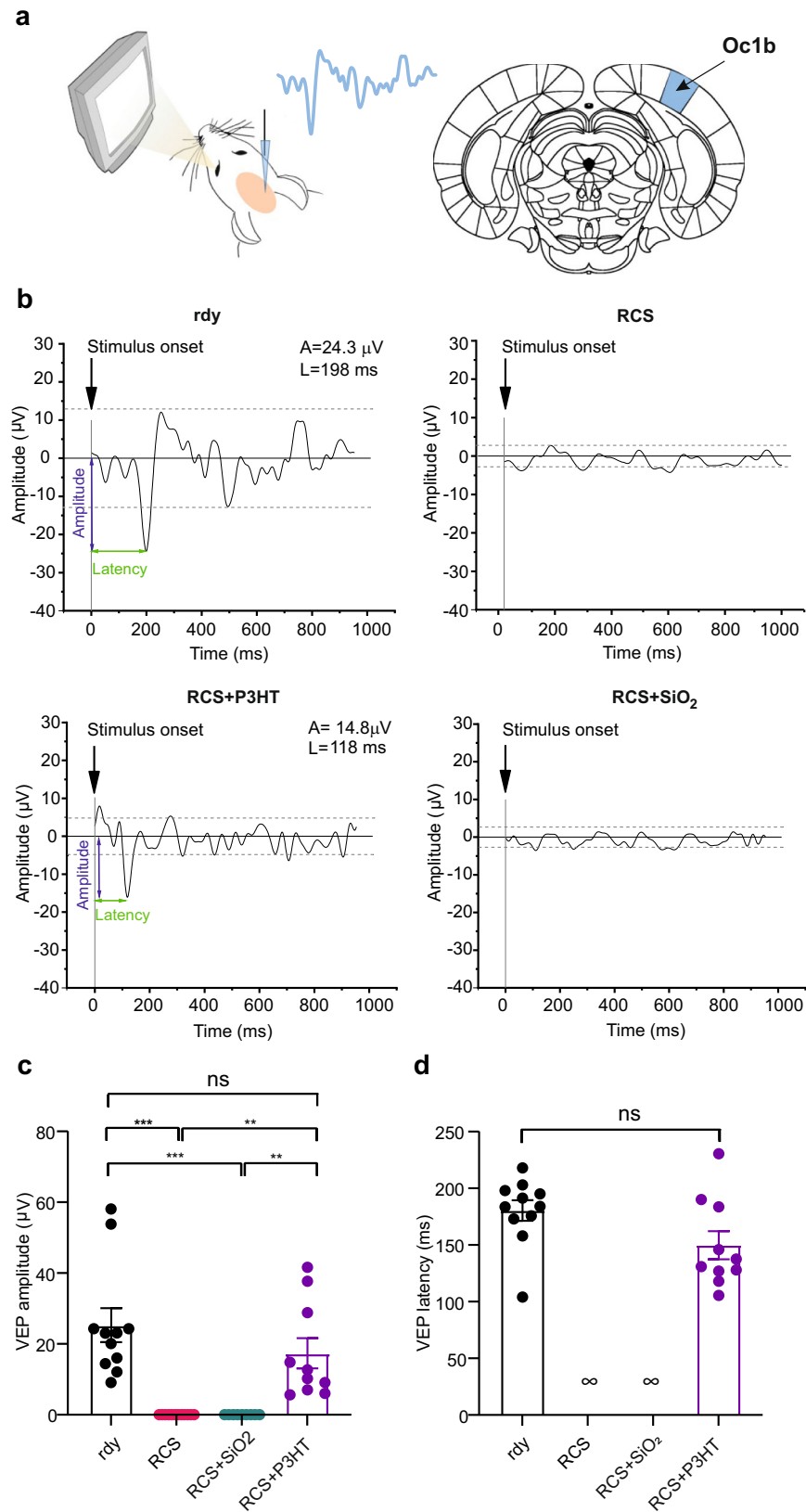

To rule out the possibility that the behavioral effects were attributable to differences in motor behavior, we evaluated the overall motor activity by computing the number of transitions between light and dark compartments. All experimental groups displayed similar motor activities (Fig. 6f), indicating that the observed behavioral effects were indeed attributable to changes in light perception. Taken together, the behavioral analysis is consistent with a sustained recovery of light sensitivity and visually driven activities by a single subretinal injection of P3HT-NPs also in the presence of a fully remodeled inner retina.

**Fig. 5 Electrophysiological rescue of visual activities in aged fully degenerate RCS retinas by P3HT-NPs. a** *Left:* Experimental setup for VEP recordings in the binocular portion of V1 (Oc1b) in response to light stimuli. *Right:* Representative VEP trace evoked by a white light (400–700 nm) flash stimulus (120 cd/m$^2$, 100 ms, 1 Hz) in an aged healthy rdy rat. The trace is the average of 200–500 sweeps. A peak detection algorithm was used to select visual evoked potentials above 2xSD of the noise (dash arrows) and analyze latency from stimulus onset and peak-to-baseline amplitude. **b** Representative VEP traces recorded in aged (13/15-month-old) healthy rdy (rdy), untreated dystrophic RCS, and RCS rats that were either injected with P3HT-NPs or sham-injected with SiO$_2$-NPs (RCS + SiO$_2$). Traces are the average of about 200–500 sweeps. A amplitude, L latency. **c** VEP amplitude in response to flash stimuli recorded in V1 at 90–150 DPI. VEP responses are totally absent in dystrophic RCS rats either untreated (RCS) or sham injected with SiO$_2$-NPs (RCS + SiO$_2$). P3HT-NPs injected dystrophic RCS rats display a significant improvement of light sensitivity, with VEP amplitudes that are not significantly different from those recorded in age-matched healthy rdy controls (one-way ANOVA/Tukey's multiple comparison tests). **d** VEP latency evoked by white light flashes at 90–150 DPI. Latencies observed in age-matched healthy rdy are not significantly different from those recorded in P3HT-NPs injected dystrophic RCS rats (Mann–Whitney's *U*-test). Data are means ± sem with individual experimental points. **p < 0.01, ***p < 0.001. Sample size: n = 11, 12, 9, and 10 for rdy, RCS, RCS + SiO$_2$ and RCS + P3HT, respectively. For exact *p*-values and source data, see Source data file.

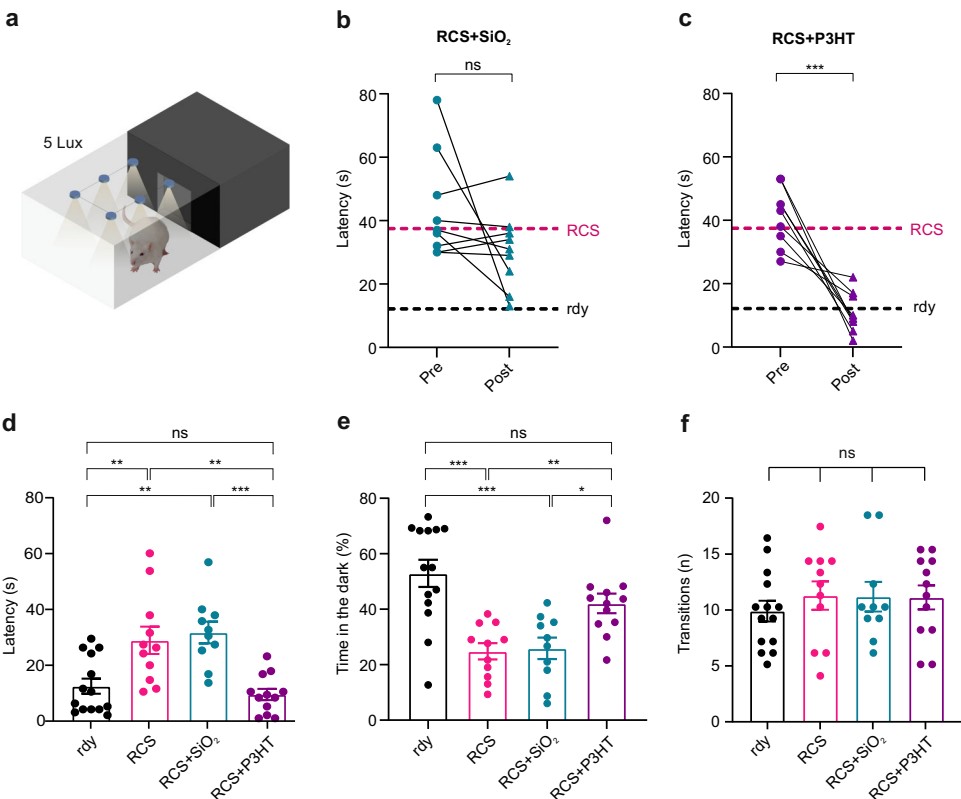

**Fig. 6 Restoration of light-escape behavior in aged dystrophic RCS rats injected with P3HT-NPs. a** Schematics of the apparatus for assessing light-driven behavior (light–dark box test). Thirty days after injection (DPI) of P3HT or SiO$_2$ (sham) NPs, aged RCS rats together with age-matched untreated RCS rats and healthy rdy controls were placed in the middle of the light compartment after 30 min of dark adaptation and irradiated with a 5-lux white source. The trial was performed 1 month before (Pre) and 1 month after (30 DPI; Post) the subretinal injection of either inert SiO$_2$- (*left*) or P3HT- (*right*) NPs by measuring the latency of escape from the illuminated side to the dark compartment, the percentage time spent in the dark compartment and the overall motor activity (number of transitions between the two compartments). **b, c** Individual pre- and post-injection light-escape latencies in RCS rats sham-injected with SiO$_2$-NPs (**b**) or injected with P3HT-NPs (**c**). The red and black horizontal lines refer to the average latency values determined in age-matched, untreated rdy controls and RCS rats, respectively. A significant reduction of light-escape latency between pre- and post-treatment was observed in aged RCS rats injected with P3HT-NPs (RCS + P3HT), as compared to sham-injected (RCS + SiO$_2$) RCS rats. Paired Student's *t*-test. Sample size: n = 13, 10, 9, and 9 for rdy, RCS, RCS + SiO$_2$ and RCS + P3HT, respectively. **d–f** The light-driven behavior of all experimental groups of 11-months-old rats was evaluated based on latency (**d**) and the percentage time spent in the dark compartment (**e**). The number of transitions between the two compartments was also monitored to check for the absence of motor impairments (**f**). P3HT-NP injected RCS rats, but not untreated or sham-injected RCS rats, display a full recovery in both escape latency and percentage time spent in the dark as compared to untreated healthy rdy controls. One-way ANOVA/Tukey's multiple comparison tests. Sample size n = 14, 11, 10, and 12 for rdy, RCS, RCS + SiO$_2$ and RCS + P3HT, respectively. Data are means ± sem with superimposed individual experimental points. *p < 0.05, **p < 0.01, ***p < 0.001. For exact *p*-values and source data, see Source data file.

**Partial restoration of visual acuity by P3HT-NPs in aged dystrophic RCS rats.** The data so far obtained demonstrated that P3HT-NPs restore light sensitivity in totally degenerated and rewired retinas. Next, we raised the challenging question of whether P3HT-NPs are also able to restore spatial vision and pattern perception notwithstanding the impaired circuitry of the inner retina.

To this aim we studied the spatial/pattern perception by investigating the optomotor response (OMR) to moving patterns of varying spatial frequency and the visual acuity based on V1

field potentials evoked by alternating patterns of increasing spatial frequency.

The OMR was evoked by moving gratings of decreasing spatial frequency that, if resolved, induce a chasing movement of the rat head and can be used as a measure of visual acuity (Fig. 7a)[51].

Old rdy rats exhibited significant OMR responses at frequencies between 0.1 and 0.2 c/deg that were totally absent in untreated or sham-injected RCS rats. Interestingly, injection of P3HT-NPs at 30 DPI restored a significant response to 0.1 c/deg, but not to 0.2 c/deg, to a level comparable to healthy rdy animals (Fig. 7a, b).

A similar picture emerged from VEP analysis in response to patterned visual stimuli of increasing spatial frequency (Fig. 7c). Aged rdy rats had values of visual acuity of about 0.5 c/deg, lower than those measured in younger animals of the same strain[13]. Strikingly, either untreated or sham-injected RCS rats were totally insensitive to the patterned stimuli independently of their spatial frequency, confirming the complete impairment of visual function. In contrast, P3HT-NP injected RCS rats had clearly detectable responses to the patterned stimuli and partially recovered pattern perception to an acuity that was about half of that of healthy rdy rats (Fig. 7d–f). Taken together, the data demonstrate the efficacy of P3HT-NPs in restoring not only light sensitivity, but also spatial vision and pattern perception in completely rewired retinas after over 1 year of photoreceptor denervation, although the extent of recovery was reduced with respect to the full restoration observed in young degenerate retinas[13].

**Reactivation of light-driven implicit memory integrated at multiple cortical levels in aged dystrophic RCS rats injected with P3HT-NPs.** To further demonstrate the restoration of the cortical processing of visual information, we subjected the same animals to a classical Pavlovian conditioning by using a mild electric foot shock as the unconditioned stimulus (US) and light stimuli as the conditioned stimulus (CS). Classical cue conditioning implies the association between US and CS with neural integration occurring at the level of higher brain centers converging on the amygdala. Light-shock pairing is known to be less effective than sound-shock since it involves more complex and indirect fear conditioning pathways than auditory-cued conditioned responses, including projections from the lateral geniculate nucleus to V1/V2 cortex, visual association area TE2, perirhinal cortex and the amygdala[52]. During the conditioning session, animals were placed in the dark chamber of the apparatus for 2-min habituation and presented with a combination of CS (light flash) and paired US (mild footshock). Following training, the animals are tested in the cue/perception test, in which the freezing behavior resulting from the presentation of the CS is scored (Fig. 8a). No differences were observed in the acquisition of a conditioned fear reaction to a light stimulus, and the freezing response recorded during each CS + US pairing progressively increased in all experimental groups (Fig. 8b). In the subsequent cued test, the light-conditioned freezing behavior increased in healthy rdy rats testifying the establishment of the implicit memory, while either untreated or SiO$_2$-NP injected rats did not show any freezing response. Notably, RCS rats injected with P3HT-NPs displayed a significantly increased light-conditioned freezing behavior, resembling healthy rdy rats (Fig. 8c). The freezing behavior recorded in the context test session, used as a negative control, did not differ among the experimental groups (Fig. 8d). Taken together, the results show that the conditioned responses (light-induced freezing), present in healthy rdy rats and totally lost in untreated or sham-injected RCS rats, are recovered up to the level of healthy controls in RCS rat which received P3HT-NPs.

**The restoration of visual activity in aged RCS rats is correlated with the density of P3HT-NPs and their contacts with second-order retinal neurons.** To further assess that the restoration of visual functions was dependent on the stimulation of the inner retina by P3HT-NPs, we conducted a multiple correlation analysis between NP density and contacts with second-order retinal neurons and the main indicators of visual performance, such as VEP amplitude, PLR and light-escape latency. We first mapped the cross-correlation between these variables (Fig. 9a) and observed a broad and significant co-linearity among all of them. Indeed, NP cluster density, number of contacts between NP microaggregates and rBPCs/HzCs and overlap of P3HT fluorescence with the specific immunoreactivity of rBPCs and HzCs were all positively correlated with VEP amplitude and extent of pupillary constriction and negatively correlated with the light-escape latency (Fig. 9a).

To observe the pattern of correlation in more detail, we performed linear regression analysis between VEP amplitude and the subretinal density of P3HT-NP clusters (Fig. 9b), their contacts and overlap (M2 coefficient) with rBPCs (Fig. 9c), as well as their contacts and overlap (M2 coefficient) with HzCs (Fig. 9d). The positive correlation was highly significant in all cases, reaching the highest statistical values for the number of contacts between P3HT-NPs and second-order retinal neurons.

We also asked if the degree of inner retina disorder and remodeling can affect visual restoration by P3HT-NPs. We correlated the main indexes of visual performance (VEP amplitude, pupillary constriction and light-escape latency) with the distance of HzCs from the RPE (Supplementary Fig. 18a) and with the rBPC misorientation (Supplementary Fig. 18b). No significant correlations were observed, supporting the central role of P3HT-NPs in the visual rescue independently of the advanced stage of the disease and the severity of remodeling of the internal retina.

## Discussion

This study demonstrates that photosensitive semiconducting P3HT-NPs can rescue visual activity in old dystrophic animals that have irreversibly lost the entirety of their photoreceptor population and have undergone a complex remodeling of the inner retina. Most of the subcortical and cortical visual responses were restored to the levels of age-matched non-dystrophic rats with the same genetic background, notwithstanding the deteriorated status of the retina and in the absence of any detectable recovery in retina morphology.

A fundamental issue to be considered in the efficacy of prosthetic interventions in RP is the progressive retina remodeling occurring during the course of the disease. Although the inner retina is not primarily involved in the degenerative process, it undergoes a complex rearrangement over time. Thus, we asked whether P3HT-NPs are still capable of rescuing visually driven activity in the presence of a destructured inner retina typical of advanced stage RP. Retina remodeling in RP is a progressive aberrant plasticity phenomenon resulting from deafferentation of the neural retina from photoreceptor inputs that leads to large-scale reorganization of the inner retina, with severe implications for the success of interventions for visual restoration[22,23]. A large body of experimental data has characterized the process of inner retina remodeling in both human RP patients[34–38] and animal models of the disease, including the Rd1, Rd10 and P23H mice[24–29], the RCS rat[30–32] and the transgenic P347L rabbit[33]. Remodeling includes neuronal cell death, neuronal and glial migration, generation of aberrant neurites and synapses and rewiring of retinal circuits, leading to changes in ganglion cells excitability and in the functional organization of receptive

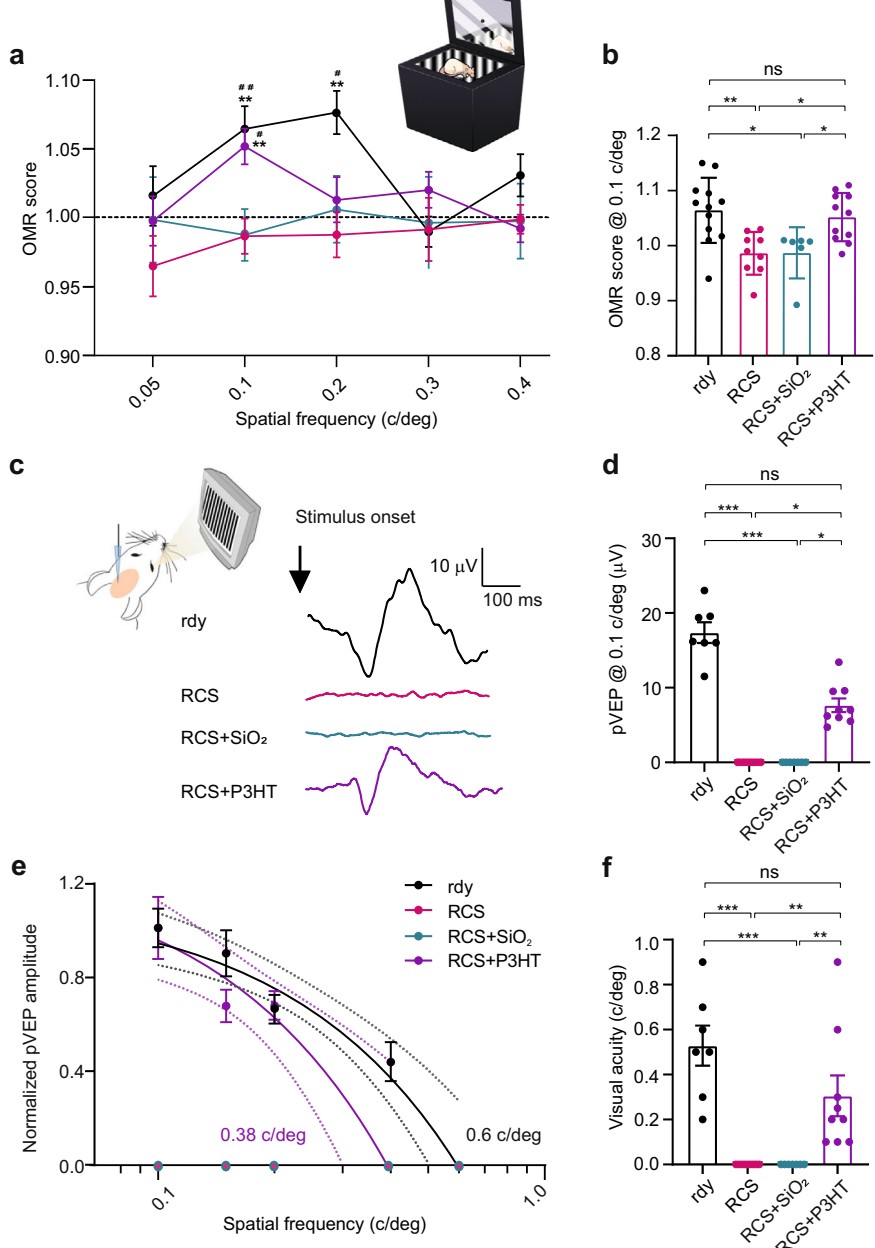

**Fig. 7 Partial restoration of visual acuity in old dystrophic RCS rats injected with P3HT-NPs. a** Optomotor response (OMR) apparatus to measure visual acuity (inset). The OMR score at the various spatial frequencies was measured at 30 DPI. The dashed line represents the cutoff of animals that did not react to a given spatial frequency. Healthy 11-month-old rdy rats displayed significant OMRs at frequencies between 0.1 and 0.2 c/deg, while age-matched, untreated and $SiO_2$-injected RCS rats did not react to the moving patterns in the 0.05–0.4 c/deg range. On the contrary, P3HT-NP injected RCS rats displayed positive responses in the 0.1–0.3 c/deg range that became significantly different from either untreated or $SiO_2$-injected rats at 0.1 c/deg. **$p < 0.01$ vs untreated RCS; #$p < 0.05$, ##$p < 0.01$ vs RCS + $SiO_2$; two-way ANOVA/Fisher's test. **b** Average OMR scores at 0.1 c/deg. P3HT-NPs injected RCS rats show no significant difference with respect to untreated rdy controls. One-way ANOVA/Holm-Sidak's test. **a**, **b** Sample size: $n = 12$, 9, 6 and 11 for rdy, RCS, RCS + $SiO_2$ and RCS + P3HT, respectively. **c** Left: Experimental setup for VEP recordings in the binocular portion of V1 (Oc1b) in response to patterned stimuli (pVEP; 120 cd/$m^2$, 100 ms, 1 Hz; spatial frequency range: 0.01–0.6 c/deg). Right: representative pVEP traces recorded in 13/15-month-old healthy rdy rats (rdy), untreated RCS rats, and RCS rats injected with either $SiO_2$-NPs (RCS + $SiO_2$) or P3HT-NPs (RCS + P3HT). **d** VEP amplitude in response to 0.1 c/deg patterned stimuli show a recovery of pVEP amplitude in RCS injected with P3HT-NPs compared to untreated or sham-injected rats. Kruskal–Wallis/Dunn's Tests. **e**, **f** Analysis of the pVEP amplitude, normalized to the response to 0.1 c/deg stimuli (**e**), and mean visual acuity, estimated as the X-intercept of the individual VEP amplitude decays (**f**), reveal a recovery in visual acuity of P3HT-NP-injected RCS rats with respect to control rdy rats. Kruskal–Wallis/Dunn's tests. Sample size in (**d**–**f**): $n = 7$, 9, 6 and 9 for rdy, RCS, RCS + $SiO_2$ and RCS + P3HT, respectively. Data are means ± sem with individual experimental points. *$p < 0.05$, **$p < 0.01$, ***$p < 0.001$. For exact $p$-values and source data, see Source data file.

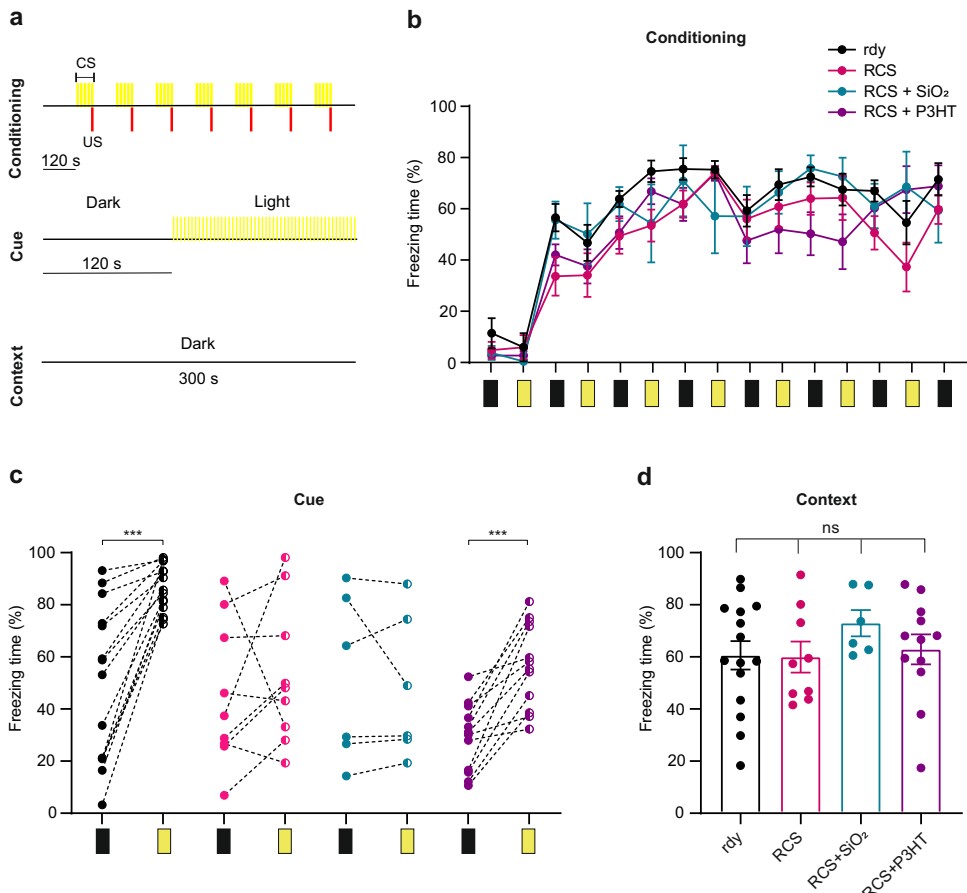

**Fig. 8 Recovery of light perception in aged RCS rats injected with P3HT-NPs as evaluated by classical conditioning. a** Schematic representation of conditioning cue and context protocols. *Top*: During the conditioning session, rats were placed in the dark chamber of the apparatus for 2-min habituation and then subjected for 9 min to seven combinations of a conditioned stimulus (CS; light flashes at 5 Hz, see Methods) paired with an unconditioned stimulus (US; mild footshock). *Middle*: During the cue test session, rats underwent 2-min habituation in the dark to explore the novel context, after which 3 min of CS were presented. *Bottom*: During the context session, rats were placed in the dark for 5 min to explore the same context of the conditioning session without the presentation of the CS-US. **b** Acquisition of a conditioned fear reaction to a light stimulus in healthy 11-month-old rdy controls and age-matched RCS rats that were untreated or subretinally injected with either P3HT-NPs or SiO₂-NPs. In the conditioning session, freezing recorded during each CS-US pairing progressively increased in all experimental groups. **c** The light-conditioned freezing behavior recorded in the cue test session increased in healthy rdy and P3HT-NP injected RCS rats, while untreated or sham-injected RCS rats did not show any freezing response. ***$p < 0.001$, paired Student's *t*-test. **d** Freezing behavior recorded in the context test session did not differ among the experimental groups. ns, not significant; one-way ANOVA/Tukey's tests. Data are means ± sem with superimposed individual experimental points. Black rectangles indicate dark time intervals; yellow rectangles represent the CS. Sample size: $n = 15$, 9, 6 and 12 for rdy, RCS, RCS + SiO₂ and RCS + P3HT, respectively. For exact *p*-values and source data, see Source data file.

fields[22,23]. Deafferented BPCs and HzCs retract most of their dendrites, HzCs develop anomalous axonal processes and dendritic stalks that enter the inner plexiform layer. RCS rats also experience these processes, although no morphological data are available. Alterations in RGC response patterns were previously reported, as well as decrements in receptive field size, impaired contrast and threshold sensitivity, shift in population profile from ON- to OFF-center RGCs and reduced RGC excitability with decreased action potential amplitude and discharge frequency[30–32].

We have recently shown that a single subretinal injection of P3HT-NPs in 3-months-old dystrophic RCS rats rescues visual performances to the levels of healthy congenic controls[13,53], indicating that, in the presence of a still fully functional neuroretina, even weak and spared retinal signals can originate light-dependent cortical responses. This phenomenon is also found in early stage RP patients who can display cortical responses to light in the subjective absence of visual perception[54,55]. The

demonstration that similar effects can be achieved in completely photoreceptor-free and destructured retinas further proves the role of P3HT-NPs as efficient nanophototransducers. Based on NP spectroscopy and electrochemistry and due to the tight and highly resistive contacts that NPs form with neuronal membranes, a mechanism can be envisaged in which, upon illumination, the negative charges accumulated at the interface capacitively affect the polarization of the neuronal membrane leading to depolarization of second-order retinal neurons[13,53]. This action is not comparable to current injection by powered photovoltaic inorganic devices or to optogenetics in which neuronal stimulation is achieved by the opening of light-gated ion channels[20]. The photoexcited NPs support space charge-induced polarization that is sensed by the environment due to the reduced dielectric screening of the cleft, thus working as a floating component in open circuit.

The over 10-months-old RCS rats used in this paper as a model of advanced stage RP display a total disappearance of retinal

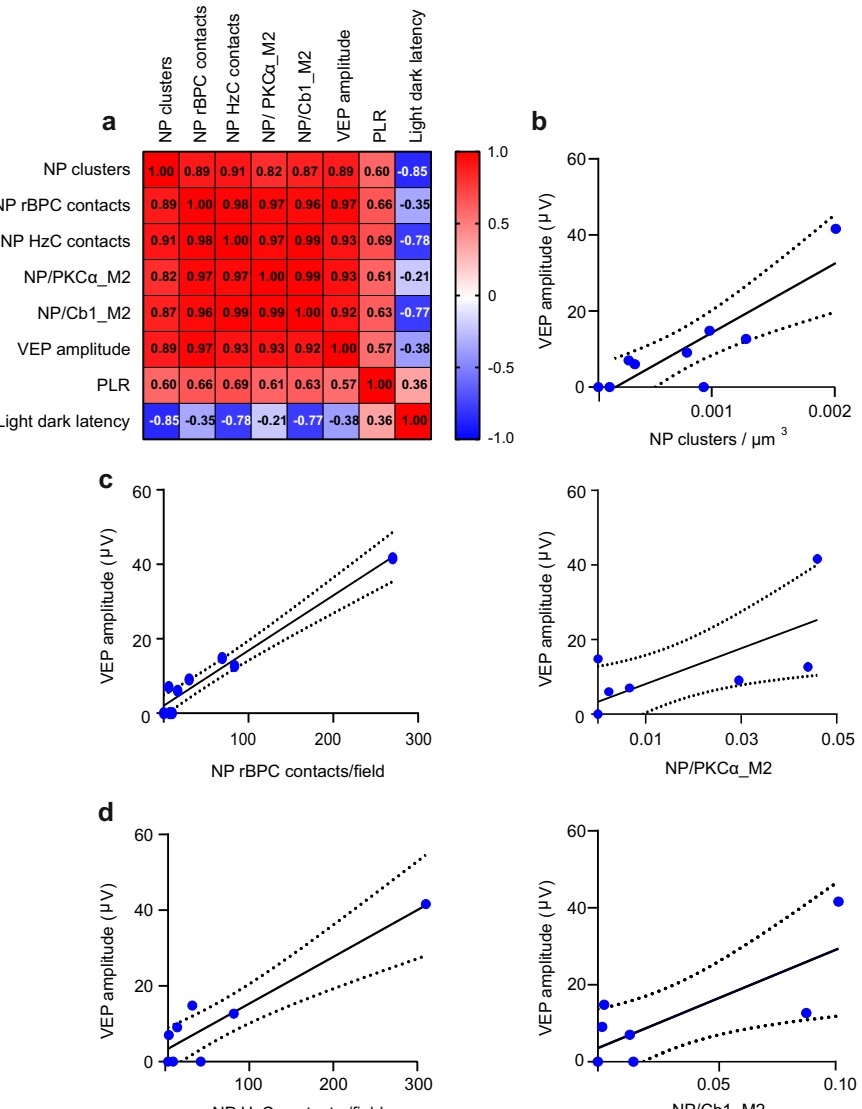

**Fig. 9 The extent of visual rescue is positively correlated with the P3HT-NP cluster density and contacts with second-order retinal neurons.**
**a** Correlation map of P3HT-NP morphological parameters in the retina and the resulting visual performances in P3HT-NP injected aged RCS rats. In the map, different variables are indicated in each row and each column, while squares represent their correlation. The Pearson's correlation coefficient, indicated in each square, shows a strong correlation between each morphological parameter (P3HT-NP clusters, contacts with rBCPs and HzC, Manders M2 overlap coefficient for PKCα-positive rBPCs and Calbindin1-positive HzCs) and visual performances (VEP amplitude, PLR constriction, light–dark escape latency). Sample size: RCS + P3HT $n = 6$. **b–d** Linear regression analysis of the correlation between VEP amplitude and morphometric parameters describing the P3HT-NP distribution (NP cluster density, rBPC/HzC contacts per field, Manders M2 coefficient for rBPC/HzC; see Fig. 3). The Pearson's correlation coefficient showed significant correlations of VEP amplitude with all the morphometric parameters that have been considered. NP cluster density: $p = 0.003$, $r = 0.7$, $n = 9$ **b**; NP-rBCP contacts/field: $p < 0.001$, $r = 0.9$, $n = 9$ and NP/PKCα_M2: $p = 0.02$, $r = 0.72$, $n = 8$ (**c**); NP-HzC contacts/field: $p = 0.0009$, $r = 0.9$, $n = 8$ and NP/Cb1_M2: $p = 0.02$, $r = 0.8$, $n = 8$ (**d**). For exact $p$-values and source data, see Source data file.

photoreceptors and cortical responses to light stimuli, in the presence of a profound disorder and remodeling of the inner retina. This experimental model is particularly interesting for assessing the translatability of the technology to the cure of RP patients, for whom prosthetic vision is only considered in advanced stages of the disease when natural vision is replaced by total blindness. Compared to early stage photoreceptor degeneration, 10-month-old RCS rats displayed complex morphological and circuit rearrangements of the inner retina, with marked displacement and misorientation of BPCs and HzCs, despite a substantially preserved inner retina molecular phenotype. Given the profound remodeling of the aged dystrophic retina, the recovery of visual activities of the RCS rats was remarkable. Light

sensitivity, evaluated from the extent and kinetics of the sub-cortical pupillary reflex, light-escape behavior and VEP amplitude were all restored to the levels of age-matched healthy rats. In addition, the cortical processing of visual information was also restored, as shown by the reappearance of the light-cued conditioned responses, a behavioral paradigm of implicit memory formation involving high order cortical areas[52]. P3HT-NPs also lead to a recovery of spatial discrimination and pattern perception as evaluated by behavioral and electrophysiological tests. These data suggest that the visual areas in the neocortex, after many months of blindness, are still capable of processing visual information, given that the animals became blind after the critical period of development of the visual cortex[56]. The only partial

recovery of spatial discrimination with respect to the complete restoration observed in retinas at an early stage of degeneration[13] is likely attributable to the complex retinal rewiring after over 1 year of photoreceptor denervation that irreversibly impairs the RGC receptive fields.

It has been reported that subretinal surgery in retinal dystrophies may have a trophic effect and increase the survival of degenerating photoreceptors[45–47]. Moreover, it has been shown that young RCS rats subjected to subretinal curettage and lavage transiently increase photoreceptor survival and light sensitivity[47]. This is not occurring in our case. We injected RCS rats at 10 months of age, a stage in which no residual photoreceptors are present, being the degeneration complete at 6 months of age or before[14,47]. Moreover, our microinjection procedure did not include subretinal lavage or extrusion of fluid or debris as done by Lorach *et al.*[47]. In our study, the possible off-target effects of surgery were always ruled out by including untreated and sham-operated groups of RCS rats. To take into account not only the effects of surgery, but also those of the permanence of foreign nanobodies in the subretinal space, the latter group underwent microinjection of inert SiO$_2$-NPs of the same size as P3HT-NPs that were found to fully mimic P3HT-NPs in terms of distribution, cluster size and retina coverage. Although the pronounced thinning of the old degenerate retinas made surgery more challenging, it did not markedly affect the distribution of P3HT-NPs with respect to what observed in younger retinas [13; this paper]: NPs remained widely distributed and highly dispersed, with the vast majority of NPs present in submicron aggregates that contacted rBPC and HzC somas and processes. Interestingly, the extent of visual restoration was solely dependent on NP density and number of contacts with second-order neurons of the degenerate retina, irrespective of the degree of inner retina rewiring.

In conclusion, the present study has shown that P3HT-NPs can be a useful strategy to restore light sensitivity and visual activities in old dystrophic animals that do not have any cortical response to light stimuli. P3HT-NPs selectively and directly restore light sensitivity by forming light-dependent "*hybrid synapses*" with second-order neurons of the neuroretina. The efficacy of P3HT-NPs, in the total absence of photoreceptors and with a massively rewired and disordered inner retina, demonstrates that retinal remodeling is not an insuperable barrier to visual restoration. Together with the extensive retina coverage, which is made possible by the NP approach, these results emphasize the high translational potential of P3HT-NPs to rescue light sensitivity in advanced and severe RP, a stage of the disease in which patients are subjected to prosthetic interventions. Moreover, the results have an important prognostic value that can be extended to prosthetic strategies other than P3HT-NPs that promote light-dependent stimulation of the inner retina.

## Methods

### Production and characterization of P3HT-NPs.
Oxidative polymerization of 3-hexyl-thiophene (3HT; Alfa Aesar, 1 g in 40 ml of CHCl$_3$) with ferric chloride (Alfa Aesar) was used to synthesize P3HT, with a procedure that displayed high reproducibility in the features of the polymer (e.g., regio-regularity, as estimated from $^1$H-NMR, dispersity and spectroscopic characteristics)[57]. H-NMR spectra were recorded at 200 MHz with a VarianVXR 200 spectrometer, using TMS ($\delta = 0.0$ ppm) as the internal reference in CDC13 solutions. The P3HT-NP preparations were obtained from the polymeric solution in organic solvent through the reprecipitation method in the absence of surfactants[40,41]. P3HT dissolved in tetrahydrofuran (Sigma-Aldrich) (8 mg/300 µl) was added dropwise to 4 ml of sterilized milliQ-water under magnetic stirring. To get rid of organic solvent traces, the P3HT suspension was dialyzed (Sigma-Aldrich dialysis sacks, 12,000 g/mol cutoff) against 2 l of sterile milliQ-water for 2 days under sterile conditions. As a control, either non-fluorescent (Microparticles GmbH) (SiO$_2$-R-L3235, 259 nm nominal diameter) or fluorescent SiO$_2$-NPs (Micromod Partikeltechnologie GmbH) (Sicastar®-redF, excitation: 569 nm, emission: 585 nm, 300 nm nominal diameter) of comparable size were used. Scanning electron microscopy (SEM) and

dynamic light scattering (DLS) were performed to characterize SiO$_2$- and P3HT-NPs. For SEM imaging, a drop of concentrated NP solution was imaged using a JEOL JSM-6490LA scanning electron microscope (JEOL Ltd, Tokyo, Japan). The size of NPs in milliQ-water (1 ml) was determined at 25 °C by the DLS technique using a 50-mW laser at 638 nm (Malvern Zetasizer NanoZS, Malvern Panalytical Ltd., Malvern, UK).

### Ethical approval and animal handling.
All manipulations and procedures involving animals were carried out in accordance with the guidelines established by the European Community Council (Directive 2014/26/EU of 4 March 2014) and were approved by the Italian Ministry of Health (Authorization # 357/2019-PR). Royal College of Surgeons (RCS) inbred dystrophic rats, together with congenic non-dystrophic (rdy) controls were kindly provided by dr. M.M. La Vail (Beckman Vision Center, University of California San Francisco, CA)[14]. Colonies were bred under standard conditions, with food *ad libitum* and on 12/12 h light dark cycle. Experimental groups were randomly selected maintaining a balance of females and males of 2 months (Young) and 13/15 months (Old) of age.

### Subretinal injection procedures.
RCS rats of either sex were anesthetized via intraperitoneal injection of diazepam (Ziapam, Ecuphar) (5 mg/kg) and subsequent intramuscular administration of xylazine (Rompun, Bayer) (5 mg/kg) and ketamine (Lobotor, Acme) (50 mg/kg), while 1% tropicamide eye drops (VISUfarma) (were administered just before the surgery start to allow for a complete dilation of the pupil). Scissors were used to perform a conjunctival dissection parallel to the *limbus* for about 2 clock hours in the superior-temporal quadrant. Subsequently, the sclera and the choroid were incised for about 0.5 mm at a distance of 1 mm from the *limbus*. Then, the retina was gently separated from the RPE close to the incision using either a small amount of viscoelastic material or the tip of surgical scissors. SiO$_2$- or P3HT-NPs (1–2 µl; 1 mg/ml) were then injected through a 38-gauge needle paying attention to penetrate the subretinal space tangential to the choroid and therefore efficiently detach the retina. Finally, diathermy (a surgical electrocautery probe, D.O.R.C) was used to coagulate the scleral incision, and the conjunctiva was repositioned over the wound. Surgical procedures were carried out preserving the sterility of the tools and of the NPs and using a Leica ophthalmic surgical microscope. The cornea was kept wet throughout the surgical procedure. After injection, the status of the retina was evaluated by indirect ophthalmoscopy. Tobramycin and dexamethasone eye drops (TobraDex 0.3% + 0.1%, Alcon) were applied for postoperative prophylaxis.

### In vivo imaging of the injected retina.
Rats were anaesthetized with an intraperitoneal injection of diazepam (5 mg/kg) followed by intramuscular administration of xylazine (5 mg/kg) and ketamine (50 mg/kg). Optical coherence tomography (OCT) was performed using a Spectralis™ HRA/OCT device. Each two-dimensional B-Scan recorded at 30° scan angle consisted of 1536 A-Scans and an average of 100 frames. Imaging was performed using the proprietary software package Eye Explorer (version 3.2.1.0).

### Retina histochemistry.
Experimental animals were euthanized by CO$_2$ inhalation and cervical dislocation. Eyes were enucleated and eye orientation was marked during processing. Eyes were fixed in 4% paraformaldehyde (Sigma-Aldrich) in 0.1 M phosphate-buffered saline (PBS, Sigma-Aldrich) for 6 h, washed in 0.1 M PBS, and cryoprotected by passing a 15–30% sucrose scale. The cornea, iris, and lens were removed and the resulting eyecups embedded in OCT freezing medium (Tissue-Tek; Qiagen), frozen in dry ice and cryo-sectioned at 50 and 25 µm using an MC5050 cryostat (Histo-Line Laboratories). Sections were mounted on gelatine-coated glass slides and stored at −20 °C before processing. We evaluated the size and the distribution of P3HT-NPs from bisbenzimide (1:300; Hoechst, Sigma-Aldrich) and antibody-stained sections by acquiring the intrinsic P3HT fluorescence ($\lambda_{ex}$, 514 nm; $\lambda_{em}$, 650–700 nm) with a Leica SP8/HyD with the super-resolution LAS-X Lightning deconvolution software (Leica, Wetzlar, Germany). For all the morphological analyses, sections were first incubated with 10% normal goat serum (NGS, Sigma-Aldrich) at room temperature for 1 h to block non-specific antibody binding, then incubated overnight at 4 °C with the primary antibody of interest at the concentrations reported in Supplementary Table 2. Alexa Fluor 488-conjugated secondary antibodies (Alexa 488-conjugated secondary antibodies hosted in goat, Thermo Fisher) were diluted 1:100 and incubated at room temperature for 1 h, let dry and mounted with Mowiol (Sigma-Aldrich). Retinal sections were imaged with a Leica SP8 confocal microscope (Wetzlar, Germany). All morphometric analyses of the retina were performed by imaging and averaging 230 × 230 × 25 µm central and peripheral z-stacks with XY resolution of 1024 × 1024 pixels, and Z steps of 700 nm, of slices passing through both the primary injection site and optic disc. Acquisition parameters were kept constant throughout the imaging sessions for comparison purposes.

*Photoreceptor counts and outer segment length.* Rods in dystrophic groups were individually counted, and the density in non-dystrophic groups measured by averaging cell counts from 3 ROIs in the ONL. This density was then used to estimate a total count per field based on the total ONL area. Cones were manually counted for all experimental groups. Where present, the length of the outer

segments (OS) of both photoreceptor types was measured at least 3 times per field using ImageJ 1.53c (NIH).

*INL cell density*. Cell density in INL was calculated by averaging nuclear counts in three separate ROIs for each image.

*Rod bipolar cell axon deviation*. The rewiring of rBPC axons was quantified by measuring their angle of intersection with a line parallel to the section's longitudinal axis, running through the midpoint of the IPL using ImageJ 1.53c (NIH). The measures were subtracted from the physiological right angle and expressed as absolute values (see Supplementary Fig. 13).

*Horizontal cell distance from RPE*. Horizontal cells were counted and their distance relative to the RPE measured individually using ImageJ 1.53c (Supplementary Fig. 13).

*Analysis of NP distribution and cell colocalization*. P3HT properties such as distribution, cluster volumes, density, and their colocalization with rod bipolar and horizontal cells, were extracted from binarized $230 \times 230 \times 50$ μm z-stack volumes stained with bisbenzimide and either anti-PKCα or anti-Cb1 antibodies, acquired at $1024 \times 1024$ resolution and imported into a custom script created in MATLAB with the Image Processing Toolbox (MathWorks). The program first reconstructed 3D volumes from the binarized z-stacks, counted the discrete objects in the P3HT intrinsic fluorescence channel and computed bounding boxes around each microaggregate to calculate their diameters. A logical "AND" operation was performed between the P3HT and antibody channels to extract only the overlapping pixels. The centroid of each discrete area of contiguous pixels was counted as a NP-Cell contact point. The M1 and M2 coefficients were obtained by dividing the total "overlap" pixel area by the total pixel area of the NP and antibody channels, respectively. The NP density was measured in a volume of interest (VOI) manually drawn around the INL. Nearest neighbor distance was calculated on binarized $290 \times 290 \times 50$ μm z-stack volumes with an xy resolution of $1024 \times 1024$ acquired from whole-mount retinas which were 3D reconstructed with the previously described script. The *pointCloud* object of the MATLAB Computer Vision Toolbox was then used to iteratively find and average the NND for each centroid.

Coverage was measured as a ratio of NP extension over total area on whole-mount retina tile scan mosaics, each tile a $1164 \times 1164 \times 130$ μm z-stack volume at $1024 \times 1024$ resolution. Total whole-mount area was measured with a manually fitted ROI, while the bounds of NP extension were measured on binarized z-max projections of the NP fluorescence using the *boundary* function in MATLAB.

*Analysis of retina inflammation*. GFAP integrated density was obtained by averaging measurements from 6 ROIs in both the IPL and ONL on a slice sum projection of each field. Sholl analysis was performed by isolating and binarizing single microglia cells. The soma diameter was used as the starting circle, ranging between 2.83 and 5.66 μm, and intersections were counted, as a function of the distance from the soma, every time a branch crossed a concentric circle of diameter increasing by 1.4 μm steps. We then calculated the cumulative sum intersections as a function of soma distance for each cell and fitted them with a Cumulative Gaussian curve to extract the mean parameter. We finally computed the cell body area by drawing an elliptical ROI on it and computing the area with the Fiji software (Supplementary Fig. 15).

*Histological staining of the retina*. Retina histology was performed using hematoxylin/eosin staining on frozen sections mounted with DPX and imaged on a light microscope.

**Total RNA extraction and real-time PCR Analysis**. Total RNA from eyecup tissue was isolated with Trizol (Invitrogen) according to the manufacturer's instructions, and reverse transcribed into cDNA using the SuperScript IV First-Strand Synthesis System (Invitrogen). Gene expression was measured by qRT-PCR using the CFX96 Touch Real-Time PCR Detection System (Biorad). Relative gene expression was determined using the $2^{-\Delta\Delta CT}$ method[58]. Gapdh and Pgk1 were used as housekeeping genes. The list of primers used is provided in Supplementary Table 3.

**Pupillary light reflex**. Following a dark adaptation of 30 min and isoflurane anesthesia (Iso-Vet, Piramal; 3% induction, 2% maintenance in oxygen), rats were positioned with the recorded eye exposed to infrared and green LEDs (780 nm and 530 nm, Thorlabs) respectively for visualization and stimulation of the pupil[13,49,50]. A camera was positioned on top of the tested eye to record the video of the pupillary reflex (Moticam 1080HDMI camera). Animals were subjected to 20 s of green light exposure at 5, 20, and 50 lux, followed by 48 s of recording in the darkness. From the recorded video, using ImageJ software, we tracked the changes in the pupil area during the experiment. Due to the variable quality of the recordings, we discarded the videos in which the pupil area measurement was not possible. From the tracking of the pupil area, we extracted six parameters: *baseline*: pupil area before the starting of the stimulus; *latency*: the interval of time between the onset of the stimulus and the start of the constriction, expressed in

milliseconds; *PLR constriction*: the averaged area of the pupil during the maximal constriction normalized to the baseline area; *PIPR dilation*: the averaged area of the pupil during the maximal dilation normalized to the baseline, following the stimulus offset; *Constriction* and *Dilation velocities*: calculated as the minimum and maximum value of the first derivative calculated during the constrictor or dilatory phase, respectively[59–62].

**In vivo electrophysiology**. Aged rats were anaesthetized with isoflurane (Iso-Vet, Piramal) (3% induction, 2% maintenance in oxygen) and placed in a stereotaxic frame (Narishige). Anesthesia level was stable during experiment and body temperature (36–37 °C) was monitored. A hole in the skull was drilled in correspondence of the binocular portion of V1 (OC1b), the dura mater was gently removed after the exposure of the brain surface. A glass micropipette (2–4 MΩ) filled with NaCl (Sigma-Aldrich) 3 M, was inserted into the visual cortex (OC1b) 4.8—5 mm from λ (intersection between the sagittal and lambdoid sutures). During the experiment, both eyes were maintained wet with saline solution (NaCl 0.9%), fixed and open with adaptable metal rings. *Full-field visually evoked potentials (ffVEPs)*. Light sensitivity was evaluated by visual evoked potentials (VEPs); refs. [13,49,63,64] recorded at about 400 μm depths in the visual cortex upon white light (400–700 nm) flashes. The visual stimuli (100 ms, 1 Hz) were generated at a luminance of 120 cd/m$^2$ (SpectroCAL MKII Spectroradiometer) by a ViSaGe MKII Stimulus Generator (Cambridge Research Systems) connected to a monitor ("20 × 22" area, 100 % brightness and contrast) located at 25 cm from the eye of the animals. *Pattern visually evoked potentials (pVEPs)*. In the case of spatial acuity, visual stimuli were horizontal sinusoidal contrast-reversing gratings of increasing spatial frequencies (0.1 to 0.6 c/deg of visual angle) at 1 Hz. Signals were amplified and band-pass filtered (0.1–100 Hz) by a NeuroLog system (Digitimer), and finally digitized thorough a multifunction I/O device from National Instruments (NI USB-6251). Signal acquisition was performed using the software MATLAB R2019b. VEPs were analyzed on averaged traces of 200–500 sweeps, in which peak detection was set above 2-fold the standard deviation (SD) of the noise (peak-to-baseline amplitude), and latency was set from stimulus onset to peak time (OriginPro2020 SR1, 9.7.0.188). Normalized pVEP amplitude to the 0.1 cycle/degree values were plotted *vs* the log of spatial frequency (0.1–1 cycle/degree) and a linear regression was computed on the decreasing pVEP amplitudes before plateauing. Visual acuity was then estimated by extrapolation of the regression line to the X-axis intercept. At the end of each session, control recordings were performed by covering the animal's eyes with black adhesive and setting luminance at 0 cd/m$^2$, to rule out electrical artefacts.

**Light–dark box test**. The light–dark test, based on the innate aversion of rodents to bright illumination, was used as a measure of light sensitivity[13,49,65]. The tests were carried out in a two-compartment box consisting of "light" and "dark" areas connected by a small door (Fig. 6a) located in a dark experimental room, the size of the "light" area being about twice the size of the "dark" compartment. After a 30 min dark adaptation, the animal was placed in the "light" area of the apparatus with the light off. Then, a video recording was performed for 5 min while the "light" compartment was illuminated with a 5-lux intensity to monitor: (i) the escape latency from the "light" area, (ii) the proportion of time spent in each compartment, and (iii) the total number of transitions between the two compartments to evaluate the light-independent motor activity. Implanted RCS rats were subjected to the behavioral test 1 month before P3HT-NPs injection (at 9 months of age), and 1 month after P3HT-NPs injection (at 11 months of age). The animals subjected to the test before the injection were removed from the box soon after the first light/dark translocation to avoid the familiarization with the experimental box and the consequent loss of novelty for the post-injection trial. Consequently, for this cohort the latency of escape was the only available parameter. The latency of non-injected RCS and rdy at the age of 9–11 months was used as reference.

**Optomotor response (OMR)**. To evaluate visual performance in each experimental group, we used the optomotor response (qOMR) as a visual-driven behavioral task. The qOMR system (Phenosys) consists of a PVC box with four screens on the inside surrounding an elevated circular platform at the center of the box and on which the animal is placed. From this location, animals can see well the visual stimulus generated on the screens. A mirror on the floor of the box creates the optical illusion of infinite depth. The system is endowed with a camera that automatically tracks body/head movements in response to shifting pattern gratings (black/white bars) of different spatial frequencies. This information is then used by the software to automatically determine visual thresholds (qOMR scores)[66]. We created and used a virtual stimulation protocol that included 0.05, 0.1, 0.2, 0.3, and 0.4 c/deg. Each spatial frequency was randomly presented for 60 s at a speed of 12° per sec. A single gray stimulus of the same duration was also included in the protocol to avoid stress in the subjects. The qOMR score is a ratio of concordance-to-discordance of body/head movements with respect to the moving gratings on the system screens that surround the animal[51]. The score of 1.0 was taken as a cut off for sensory perception: animals with qOMR scores < 1 are considered not to perceive the spatial frequency under investigation. Plotting qOMR scores versus

spatial frequencies is an effective behavioral strategy to assess visual discrimination abilities in each experimental animal.

**Light-cued classical conditioning**. The classical conditioning paradigm was performed in an environmental chamber equipped with a grid to deliver the shock (Med Associated Inc.). A camera mounted on the front door recorded test sessions, which were automatically scored using the integrated software for the identification and quantification of behavioral freezing. The adopted protocol was composed of 3 phases (see Fig. 8a)[67]:

 I. The Conditioning session consisted of 2 min of habituation in which animals freely moved to explore the environment. Immediately after, a sequence of seven repetitions of white light flashes (22 lux at 5 Hz; 2 s on / 2 s off repeated for 5 times) serving as the conditioning stimulus (CS) with 60 s off between each CS. During the last 2 s light on, a mild foot shock (0.5 mA) was delivered as the unconditioned stimulus (US). Each mouse received seven CS-US pairings separated by intervals of variable duration[67]. To evaluate the US-CS associative learning, we quantified the freezing behavior during each US-CS pairing.

 II. The Cue test session took place the day after the conditioning session in the same apparatus. To avoid chamber-US association and only evaluate the CS-US association, the environment was altered by covering the grid floor with a smooth white plastic sheet and replacing the arena with black and white stripped walls. In addition, a new aromatic odor (Vanillin, Sigma-Aldrich) and two wired cups were introduced in the chamber. After 5 min habituation to the new chamber, the test began. After 2 min in the dark without stimulation, each animal received 3 min of continuous CS. To evaluate if rats were able to perceive the light stimulus, the freezing behavior was quantified during the 2 min in the dark and during the 3 min of CS.

 III. The Context session took place the day after the Cue test session in the same chamber as the conditioning session. Each rat was placed in the chamber for 5 min in the absence of CS and US, during which freezing behavior was scored.

Before each test session, rats were dark-adapted for 1 h and all experiments were conducted in the dark. For each session of the test, the time percentage of freezing behavior was expressed as freezing time (s)/total time (s) of the session.

**Statistical analysis**. The sample size needed for the planned experiments ($n$) was predetermined using the G*Power software for ANOVA test with four experimental groups, considering an effect size = 0.25–0.40 with alpha (type-I error) = 0.05 and 1-ß (type-II error) = 0.9, based on similar experiments and preliminary data[11]. Experimental data are expressed as means ± sem throughout, with $n$ as the number of independent animals ($n$). Normal distribution was assessed using D'Agostino-Pearson's normality test. To compare two normally distributed sample groups, either Student's $t$-test or Mann–Whitney $U$-test was used. To compare more than two normally distributed sample groups: one-way ANOVA followed by Tukey's test or Holm-Šídák multiple comparison test, mixed-effects model with Geisser-Greenhouse correction followed by Holm-Šídák multiple comparison test or two-way ANOVA/Fisher's test. To compare more than two non-normally distributed sample groups, non-parametric Kruskal–Wallis followed by Dunn's multiple comparison test was used. Correlation tests between variables were performed based on Pearson's correlation coefficient. Statistical analysis was carried out using OriginPro2020 SR1, MATLAB R2019b, and GraphPad Prism 6.07 & 9.

**Reporting summary**. Further information on research design is available in the Nature Research Reporting Summary linked to this article.

## Data availability

All relevant data supporting the key findings of this study are available within the article and its Supplementary Information files or from the corresponding author upon reasonable request. Source data are provided with this paper.

## Code availability

Scripts used for morphometric analyses are accessible at https://doi.org/10.5281/zenodo.6577009.

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

## Acknowledgements

The Authors thank dr. Fabio Piazza (ENT Department, ASST Mantova, Mantua, Italy) for critical reading of the manuscript and enlighting discussions; M.M. La Vail (Beckman Vision Center, University of California San Francisco, CA) for kindly providing non-dystrophic RCS-rdy+ and dystrophic RCS rats; dr. A. Russo (Department of Ophthalmology, Sacro Cuore Don Calabria Hospital, Negrar, Italy) for assistance in the subretinal microinjections; drs. M. Cilli and L. Emionite (IRCCS Ospedale Policlinico San Martino, Genova, Italy) for assistance in the surgical procedures; R. Ciancio, I. Dallorto, A. Mehilli, R. Navone and D. Moruzzo (Istituto Italiano di Tecnologia, Genova, Italy) for technical assistance. The research was supported by Fondazione Cariplo (project "Nanospark" 2018-0505), H2020-MSCA-ITN 2019 "Entrain Vision" (project 861423), EuroNanoMed3 (project "Nanolight" 2019-132), IRCCS Ospedale Policlinico San Martino (Ricerca Corrente and 5 × 1000 grants) and the Ra.Mo. Foundation (Milano, Italy). The Fondazione 13 Marzo (Parma, Italy) is gratefully acknowledged for the constant support to the project.

## Author contributions

G.M. and S.P. fabricated and characterized the NPs under the supervision of G.L.; G.P., M.M., and M.A. developed and executed the subretinal microinjections; M.M., S.D.M. S.F., and E.C. performed the OCT experiments; D.S. and S.C. performed histological analyses under the supervision of S.D.M.; S.D.M., and E.C. performed and analyzed the pupillary reflex experiments; S.F. performed and analyzed the in vivo electrophysiology experiments with the help of E.C., S.B., and J.F.M.-V.; G.M. and J.F.M.-V. performed and analyzed the OMR experiments; G.M., S.F., and C.M. performed and analyzed the classical conditioning experiments; M.D.F. and S.F. performed and analyzed the behavioral tests; A.R. performed qRT-PCR; G.C. and R.S. developed and implemented the mathematical model and performed the numerical simulations; F.B., G.P., and G.L., conceived and supervised the project; F.B. wrote the manuscript; all authors discussed the experimental results and revised the manuscript.

## Competing interests

The P3HT-NP technology is the subject of the US patent application US 16/005,248 "*Eye-injectable polymeric nanoparticles and method of use therefor*" by Istituto Italiano di Tecnologia and Ospedale Sacro Cuore Don Calabria that was submitted June 11, 2018, and is currently licensed to Novavido s.r.l.—a company that develops organic retinal prostheses. The Authors declare the following competing interests: F.B., G.L. and G.P. are inventors in the patent application and cofounders of Novavido s.r.l.; F.B. and G.L. are scientific consultants of Novavido s.r.l. The other authors declare no competing interests.
