## [Peer Review File · Nature Communications]

Light-induced charge generation in polymeric nanoparticles restores vision in advanced-stage retinitis pigmentosa ratsREVIEWER COMMENTS

Reviewer #1 (Remarks to the Author):

Manuscript: "LIGHT-INDUCED CHARGE GENERATION IN POLYMERIC NANOPARTICLES RESTORES VISION IN ADVANCED-STAGE RETINITIS PIGMENTOSA RATS" by Francia et al.

General Points:

This is an interesting manuscript analyzing the effects of conjugated polymer nanoparticles on physiological signals in the optic nerve and visually driven activities when microinjected in 10- 40 months-old RCS rats bearing fully light-insensitive retinas with heavily remodeled inner layers. The authors conclude that the extent of visual restoration positively correlated with the nanoparticle density and hybrid contacts with second-order retinal neurons. The potential significance of these findings is very high as there is an unmet need of therapies that can recover and/or improve vision in retinal degenerations. However, there several issues that need to be addressed by this study to improve the impact of the findings.

Specific Points:

- The text should be revised and edited for clarity.
- In Figure 1a, the authors show a very high magnification of OCT images of young and old RCS rat model. The images are very high magnification; it would be helpful to have the normal view of the OCT images, with an insert in higher magnification. Also, the dashed lines are helpful to highlight the retinal layers but when over the whole image, it affects the information; perhaps the dots could be added only to part of the figure.
- The images in Supplementary Figure 2 are again of high magnification; it would be helpful to have the normal view of the OCT images, with an insert in higher magnification.
- The images in Supplementary Figure 2a are not convincing; nuclei staining is very weak. The images need to be improved as information is needed to locate the NPs in the injected eyes. Why is the RCS+P3HT Old NPs signal so weak (this is the only figure of the panel with nuclei well labeled)? It looks as if there are less P3HT in the RCS rats. Is that real? What is the cause of this difference?
- The authors write that there was a significant overlap between the P3HT fluorescence and the immunoreactivity for BPCs and HzCs. Fig. 2b and 2c do not show a high degree of overlap between these markers. Please comment.

- The authors report degeneration-induced inner retinal rewiring is not modified by the injection of P3HT-NPs and does not affect P3HT-NP-dependent visual rescue (Supplementary Figure 6). This data would be stronger if the authors provided images of the rats not injected, injected with SiO₂ and injected with P3HT-NPs and labeled with BPCs and HzCs markers.
- The tracing of the pupil with PLR and PIPR in Figure 3 and Supplementary Figure 8 affects the image and make it difficult to realize the magnitude of the changes.
- The text describing Figure 4 (page 9, lines 265-277) does not seem to describe all the changes observed; it also cites young and aged animals not depicted in the figure. Please edit the text.
- The Text described Figure 5d, 5e, 5f but the figure is not labeled. Please edit it. The text also states that aged P3HT-NP-injected dystrophic RCS rats significantly recovered their light-driven behavior to the levels of non-injected rdy controls. However, that is not represented in the graphs presented: all the changes were ns between rdy and RCS+P3HT in the graph. Please check it.
- One important issue not addressed in this study is the use of albino rats. The significance of the findings presented here would be higher if some of the key findings were also analyzed in the retinas of pigmented animals.
- The authors stated that GFAP and Iba-1 staining were not significantly affected. However, the images presented did show changes for example in the shape of the Iba-1 cells and the distribution of GFAP. The authors should discuss these changes.

Reviewer #2 (Remarks to the Author):

The authors built on former results indicating that P3HT-NPs restore visual function in a rat model of retinitis pigmentosa. In the current work, the authors changed to an even more challenging protocol and injected P3HT-NPs subretinally to old (age 10 months) RCS rats which had been blind already for many months. As in perspective, a treatment of patients would be rather applied at an end-stage of the disease, the chosen protocol gives meaningful results also for evaluating translational possibilities. In addition, the authors also provide in-depth analysis of the pathophysiology, the possible effects of the treatment and the underlying mechanisms.

The results show a significant effect regarding vision restoration which is demonstrated on physiological and behavioural level. Interestingly, cellular, molecular, immunohistochemical analysis and mathematical modelling indicate that the effects are based indeed on the charge generation of the P3HT-NPs and not on protection or regeneration of retinal cells or networks of retinal cells.

In summary, not only the demonstrated results are very promising, also the scientific analysis provides an in depths understanding of the mechanisms.

There are some issues which may be considered:

General:

The authors demonstrated impressively by physiological and electrophysiological in vivo experiments and behavioural tests that visual function is restored. However, there are additional questions which are of interest and which may be addressed in the discussion:

- After many months of blindness, the area of the visual cortex may have been changed due to post-lesional plasticity. Do the authors expect that after P3HT-NPs injection the rats need to re-learn seeing? Will this change again the visual cortical area?
- Do the P3HT-NPs-treated rats also restore spatial vision and pattern perception with such late-stage treatment?

Detail:

- Supplementary Figure 8 d, e and f is confusing as the green lines (SiO₂) are similar to the black lines (rdy control animals) and P3HT-NPs values are similar to RSC values. This is also in contrast to the explanations in the legend. Please correct or explain.
- The authors show distribution of the P3HT-NPs in Figure 2 and Supplementary Figure 3. However, these images show rather restricted areas of the retina. How is the “global” distribution of the P3HT-NPs? Are they clustered around the injection site or are they distributed more evenly across the retina? Did the authors analyse the distribution of the NPs in a retina whole-mount? Or they may refer to former results

Minor:

- The authors may consider to transfer Supplementary Figure 4 to the main text.
- Although the authors write it the title “... restores vision...”, in the manuscript they use in general “rescuing” vision. However, what is gone (rats are blind at 10 months) can not be “rescued”, rather “restored”
- The authors used male and female rats for their experiments which increases the meaning of their experiments. However, is there anything known if there are differences in the pathology of male and female RCS rats?
- „.....restoring visual activities in fully degenerate retinas in the absence of residual photoreceptors and in the presence of an intense inner retina rewiring...“

Comment: photoreceptors are fully degenerated, retina is not fully degenerated

- “...aversion of nocturnal rodents to illuminated areas...” Comment: although rats have an aversion to illuminated areas, they are not nocturnal animals
- Do the authors expect any effects if P3HT-NPs are injected intravitreally? This would be easier, especially for clinical applications

Reviewer #3 (Remarks to the Author):

This manuscript presents an investigation with conjugated polymer nanoparticles possessing intrinsic light-induced charge generation that allows for reinstating of physiological signals and visually driven activities. The work was performed in 10 months old RCS rats and this is very pertinent, as these old animals quite faithfully represent advance stages of retinal degeneration, with light-insensitive retinas and remodeled inner layers. This work is also a continuum of the previous work of the group (Maya-Vetencourt, J. F. et al. *Nature Nanotechnology* 15,698– 708, 2020) that evaluated the effects of the conjugated polymer nanoparticles P3HT NPs in 4- and 11-months old RCS rats. Here, the Authors found that P3HT-NPs distribute evenly in the subretinal space of young and old RCS rats, contact bipolar and horizontal cells after a single subretinal injection and lead to restoration of visually evoked potentials in the primary visual cortex and pupillary constriction light-driven behavioral responses (latency of escape from the illuminated area to darkness and percentage of time spent in the dark), and claimed that the extent of visual restoration positively correlated with the nanoparticle density and hybrid contacts with second-order retinal neurons.

Questions, concerns and suggestions:

1. There are no histological data to compare the status of the P3HT-particle injected animals and the glass-particle injected animals. Therefore, it is not possible to be sure that the surgeries went similarly in both cases. Quantification of the particle distribution is needed to ensure that the surgeries were identical for the same animals as those used in the physiological experiments.
2. The surgery in RCS rats was found to clean some accumulation of cell debris resulting in improvement of visual function (Lorach et al., *Sci Rep.* 2018; 8:11312). It is therefore important to evaluate how the surgery has removed all these debris and demonstrate that the observed effect is not due to this surgical effect. This question relates to point 1 and to the importance of verifying that the particle injections were identical with P3HT- and glass-particles.
3. A major question concerns the differences between the efficient light intensity in vitro (> 10mW/mm²) and in vivo (5 lux). The authors should explain this discrepancy with some experimental data, otherwise, it supports the hypothesis explained in point 2 with an improved survival following the surgery. What is the rationale for the used light intensities: 5-50 lux for pupillary light reflex studies; 5 lux for light-dark box test. Please provide robust evidence about the efficacy at 5-lux conditions on the activation of the nanoparticles.
4. In the previous paper in *Nature Nanotechnology*, the key element appeared to be the infrared sensitivity. Why this key factor was not used here?
5. The remodeling is not so major at this age (10 months old RCS rats) and it is possible that these animals still have some light perception as indicated by the response of the control animals in the light dark test.
6. Throughout the manuscript, the Authors talk about "restoring visual activities". They should clearly say which of the visual functions exactly have been restored. From the results, it appears that no visual

function is restored except light perception! Visual perception is related to pattern perception and no data can clearly demonstrate pattern perception in this paper.

7. The link to the WHO is not the most appropriate reference for visual impairment due to inherited retinal diseases.

8. on line 69, the Authors say that "Overall, this approach is rather cumbersome and has severe limitations in performance" talking about optogenetics and without providing a references for this statement. References measuring the theoretical visual acuity with optogenetics should be cited.

9. on lines 165-166 "P3HT-NPs remained confined to the outer retina, replacing the lost photoreceptors, with no tendency to radially diffuse to the inner retina layers": please explain how/why this distribution is so selective? is it possible to generate pattern perception with so few particles?

10. lines 219-223 "Photoreceptor markers were undetectable in all aged RCS rats, while the inner retinal markers were not affected by the microinjection of either P3HT-NPs or SiO₂-NPs". The Authors insist on the fact that the photoreceptors were no more there, so this statement sounds bizarre, please discuss this, possibly in the light of the photoreceptor survival demonstrated by Lorach et al. following a subretinal surgery because it washes out the cell debris below the retina.

11. at different places throughout the manuscript the Authors talk about "visual rescue". First, the term "vision restoration" could better fit the effect they intend to describe; second, there is no demonstration of vision restoration but improved light perception. No temporal and spatial resolution.

12. lines 332-333 "Light stimuli, through P3HT-NPs, activated a correlated spiking activity in the optic nerve subserving a variety of functions": There is no measurement of spiking activity but only of visually evoked potentials. How can the Authors assert a variety of functions apart from light perception? There is no definition of spatial resolution neither temporal resolution.

13. in the discussion, when talking about "affecting the polarization" and commenting optogenetics, the Authors should also mention the mediation of depolarization with channelrhodopsins or hyperpolarization with halorhodopsins

14. a big part of the discussion focuses on the previous data, ref 11, instead of the data from this study

REVIEWER COMMENTS

Reviewer #1

General Points:

This is an interesting manuscript analyzing the effects of conjugated polymer nanoparticles on physiological signals in the optic nerve and visually driven activities when microinjected in 10- 40 months-old RCS rats bearing fully light-insensitive retinas with heavily remodeled inner layers. The authors conclude that the extent of visual restoration positively correlated with the nanoparticle density and hybrid contacts with second-order retinal neurons. The potential significance of these findings is very high as there is an unmet need of therapies that can recover and/or improve vision in retinal degenerations. However, there several issues that need to be addressed by this study to improve the impact of the findings.

We thank the Reviewer for the positive comments. We addressed the raised issues and are grateful for the opportunity to improve the quality of the work.

Specific Points:

- The text should be revised and edited for clarity.

We fully revised and edited the text as suggested.

- In Figure 1a, the authors show a very high magnification of OCT images of young and old RCS rat model. The images are very high magnification; it would be helpful to have the normal view of the OCT images, with an insert in higher magnification. Also, the dashed lines are helpful to highlight the retinal layers but when over the whole image, it affects the information; perhaps the dots could be added only to part of the figure.

In **Figure 1a**, we used high magnification to render the rat retina layers clearly identifiable. On the other hand, being Figure 1 extremely complex, we are inclined to keep the high magnification only (partially removing the dashed lines, as suggested) and provide the corresponding normal views of the OCT images in the **new Supplementary Figure 1**.

- The images in Supplementary Figure 2 are again of high magnification; it would be helpful to have the normal view of the OCT images, with an insert in higher magnification.

As suggested, we now provide normal views of the OCT images of the former Supplementary Figure 2 (now **Supplementary Figure 3**), together with high-mag OCT and fundus images for the three retinal regions analyzed (temporal, optic nerve and nasal) in injected rats.

- The images in Supplementary Figure 3a are not convincing; nuclei staining is very weak. The images need to be improved as information is needed to locate the NPs in the injected eyes. Why is the RCS+P3HT Old NPs signal so weak (this is the only figure of the panel with nuclei well labeled)? It looks as if there are less P3HT in the RCS rats. Is that real? What is the cause of this difference?

We agree that the images in Supplementary Figure 3 were not representative and had some label intensity problems. Apart from those, the lower density of the P3HT-NP in the dystrophic retinas with respect to healthy animals was attributable to the higher diffusion in the subretinal space due to the absence of photoreceptors. However, we thought that the injection of NPs in the subretinal space of normal animals, in the presence of photoreceptors, is not so meaningful and does not contribute to the theme of the paper. Then, for clarity, we decided to leave it out and concentrate on the distribution of P3HT- and control SiO₂-NPs (required by Reviewer #3, point 1) in dystrophic retinas only, by giving large scale images of whole mount retinas, high magnification images and a thorough quantification of retina coverage and mean nearest neighbor distance between NPs. These data compose now the new main **Figure 3** and new **Supplementary Figure 4**.

- The authors write that there was a significant overlap between the P3HT fluorescence and the immunoreactivity for BPCs and HzCs. Fig. 2b and 2c do not show a high degree of overlap between these markers. Please comment.

With the confocal analysis we barely resolve single NPs. Thus, the analysis refers to NP clusters whose size distribution was analyzed in Supplementary Figure 3 in the previous version (now **new Figure 3**). We extended our morphological analysis by providing the Manders M1 coefficient (percentage of P3HT fluorescence overlapped with BPC/HzC staining). It should be considered that the P3HT/neuron overlap is also limited by the fact that we only labeled rod BPCs and HzCs, which do not represent the entirety of second order neurons. In fact, it has been reported that in the mouse, rod BPCs only represent the 27% of all BPCs (Wässle *et al.*, 2002; Strettoi & Volpini, 2002). Based on this consideration, we are now also providing an analysis of the density of NP clusters in the INL volume.

- The authors report degeneration-induced inner retinal rewiring is not modified by the injection of P3HT-NPs and does not affect P3HT-NP dependent visual rescue (Supplementary Figure 6). This data would be stronger if the authors provided images of the rats not injected, injected with SiO₂ and injected with P3HT-NPs and labeled with BPCs and HzCs markers.

As suggested, we have added representative images of the 4 experimental groups to the former **Supplementary Figure 6**.

- The tracing of the pupil with PLR and PIPR in Figure 3 and Supplementary Figure 8 affects the image and make it difficult to realize the magnitude of the changes.

We have modified the thickness of pupil tracing in both figures (now **new Figure 5** and **new Supplementary Figure 10**). We hope it will not interfere with the detection of pupil changes anymore.

- The text describing Figure 4 (page 9, lines 265-277) does not seem to describe all the changes observed; it also cites young and aged animals not depicted in the figure. Please edit the text.

We removed the sentence and strictly referred to the data shown in the figure (now **new Figure 6**).

- The Text described Figure 5d, 5e, 5f but the figure is not labeled. Please edit it. The text also states that aged P3HT-NP-injected dystrophic RCS rats significantly recovered their light-driven behavior to the levels of non-injected rdy controls. However, that is not represented in the graphs presented: all the changes were ns between rdy and RCS+P3HT in the graph. Please check it.

We apologize for the lack of panel letters d, e, f. The figure (**new Figure 7**) is now correctly labeled. The fact that all the changes between rdy and RCS+P3HT were not significant exactly demonstrate the full recovery of the visual performance in P3HT-injected blind rats.

- One important issue not addressed in this study is the use of albino rats. The significance of the findings presented here would be higher if some of the key findings were also analyzed in the retinas of pigmented animals.

We do not fully understand, at the best of our knowledge, how pigmentation can affect the results. By the way, the RCS rat strain that we used is "pink-eyed", and not a full albino phenotype, being slightly pigmented in the hair. Moreover, in the fully pigmented RCS rat strain, pigmentation has been demonstrated to slightly shift the photoreceptor degeneration curves to the right, moderately slowing down the rate of photoreceptor death. However, the degeneration has long been completed at the age we tested our animals, irrespective of pigmentation. This was clearly shown by Matthew M. LaVail (UCSF, San Francisco, CA), who created the RCS strains and distributed them to most world laboratories. According to the UCSF document (data published in LaVail & Battelle, *Exp. Eye Res.* 21: 167-192, 1975 and reported below), at 3 to 4 months of age in the pink-eyed RCS rat, photoreceptors are gone, in both pigmented and pink-eyed RCS rats.

Posterior (Central) Retina of Pink-eyed RCS and Pigmented RCS-p⁺ Rats

Peripheral Retina of Pink-eyed RCS and Pigmented RCS-p⁺ Rats

Repeating the same experiments in the pigmented RCS strain is troublesome. In fact, it would imply a new application to the ethical committee and Italian Ministry of Health for the authorization to breed a new genotype of animal (about 6 months), get breeding couples and build up the colony (about 12 months), aging of the animals (12 months) to end up with surgery and physiological follow-up (3-6 months). All together a 2/3-year time that is not compatible with the *Nat. Comm.* editorial rules and timing of publication.

- The authors stated that GFAP and Iba-1 staining were not significantly affected. However, the images presented did show changes for example in the shape of the Iba-1 cells and the distribution of GFAP. The authors should discuss these changes.

We integrated the data shown in the previous Supplementary Figure 7 (now **new Supplementary Figure 8**) with the quantification of the microglial shape using Sholl analysis. Regarding GFAP, there is indeed a trend for an attenuation of the astrogliosis in rats injected subretinally, particularly in the case of P3HT-NPs. However, the effects on GFAP expression, as well as the correlation between P3HT-NP density and GFAP attenuation were non-significant.

Reviewer #2

The authors built on former results indicating that P3HT-NPs restore visual function in a rat model of retinitis pigmentosa. In the current work, the authors changed to an even more challenging protocol and injected P3HT-NPs subretinally to old (age 10 months) RCS rats which had been blind already for many months. As in perspective, a treatment of patients would be rather applied at an end-stage of the disease, the chosen protocol gives meaningful results also for evaluating translational possibilities. In addition, the authors also provide in-depth analysis of the pathophysiology, the possible effects of the treatment and the underlying mechanisms. The results show a significant effect regarding vision restoration which is demonstrated on physiological and behavioral level. Interestingly, cellular, molecular, immunohistochemical analysis and mathematical modelling indicate that the effects are based indeed on the charge generation of the P3HT-NPs and not on protection or regeneration of retinal cells or networks of retinal cells. In summary, not only the demonstrated results are very promising, also the scientific analysis provides an in depths understanding of the mechanisms.

We thank the Reviewer for her/his positive evaluation and interpretation of our work.

There are some issues which may be considered:

General:

The authors demonstrated impressively by physiological and electrophysiological in vivo experiments and behavioural tests that visual function is restored. However, there are additional questions which are of interest and which may be addressed in the discussion:

- After many months of blindness, the area of the visual cortex may have been changed due to post-lesional plasticity. Do the authors expect that after P3HT-NPs injection the rats need to re-learn seeing? Will this change again the visual cortical area?

A very fascinating suggestion that we now mention in the Discussion (p. 14). Based on previous studies on visual cortical plasticity, these animals that become blind after the end of the critical period should have a relatively normal visual cortex development and retain it over the blindness period. It will certainly be an interesting topic for future investigations.

- Do the P3HT-NPs-treated rats also restore spatial vision and pattern perception with such late-stage treatment?

Another very interesting question that we tried to answer using a new cohort of old RCS rats subretinally injected with either P3HT or SiO₂ NPs. We studied the spatial/pattern perception by investigating the optomotor response (OMR) to moving patterns at changing spatial frequency and the visual acuity based on the V1 field potentials evoked by alternating patterns of increasing spatial frequency. The results are shown in the **new Figure 8** and described in the Results (pp. 10-11).

The OMR, based on head tracking, was generally weak in old healthy rdy rats, with significant responses at frequencies between 0.1 and 0.2 c/deg that were totally absent in untreated or sham-injected RCS rats. Interestingly, injection of P3HT-NPs rescued the response to 0.1 c/deg to a level comparable to healthy rdy animals. A similar picture emerged from the pattern VEP analysis. While untreated or sham-injected RCS rats were totally insensitive to the patterned stimuli independently of their spatial frequency, P3HT-NP injected RCS rats partially recovered pattern perception with a resulting acuity about half of that determined in healthy rdy rats, as compared to untreated or sham-treated RCS rats which did not exhibit any response to spatial patterns. Importantly, the data demonstrate the efficacy of polymeric NPs in restoring, not only light sensitivity, but also spatial vision and pattern perception. Moreover, the data also suggest that, in the presence of completely rewired retinas, after over one year of photoreceptor denervation, the recovery of the visual activities is reduced with respect to the one observed in “freshly” degenerate retinas (Maya-Vetencourt *et al.*, *Nat Nano* 2020).

To further prove the cortical processing of visual information, we subjected the same animals to a classical Pavlovian conditioning by using a mild electric shock as the unconditioned stimulus (US) and a train of light flashes as the conditioned stimulus (CS). Classical cue conditioning implies the association between US and CS with neural integration occurring at the level of higher brain centers, including the visual cortex, the perirhinal cortex, and the amygdala (Newton *et al.*, 2004). The results,

reported in the **new Figure 9** and on pp. 11-12, show that the conditioned responses (light-induced freezing), present in healthy rdy rats and totally lost in untreated or sham-injected RCS rats, are recovered up to the level of healthy controls in RCS rat which received P3HT-NPs.

Detail:

- Supplementary Figure 8 d, e and f are confusing as the green lines (SiO₂) are similar to the black lines (rdy control animals) and P3HT-NPs values are similar to RCS values. This is also in contrast to the explanations in the legend. Please correct or explain.

We really apologize for the mislabeling of panels d,e,f in the old Supplementary Figure 8. It was a trivial mistake that we now corrected in the **new Supplementary Figure 9**.

- The authors show distribution of the P3HT-NPs in Figure 2 and Supplementary Figure 3. However, these images show rather restricted areas of the retina. How is the “global” distribution of the P3HT-NPs? Are they clustered around the injection site or are they distributed more evenly across the retina? Did the authors analyse the distribution of the NPs in a retina whole-mount? Or they may refer to former results

Supplementary Figure 3a was indeed taken from whole mount retinas. We have now analyzed whole mount at low and high magnification in retinas injected with either P3HT or fluorescent SiO₂ NPs, and provide quantification of cluster size distribution, retina coverage by NPs and nearest neighbor distance. The retina-wide distribution of both P3HT and SiO₂ NPs shows that the highest NP density is present around the injection site, but that NPs can reach also distant areas of the retina. The figure has now become main **new Figure 3**.

Minor:

- The authors may consider transferring Supplementary Figure 4 to the main text.

Thank you for the suggestion. It is certainly useful to find a graphical representation of the experimental protocol in the main figures. The old Supplementary Figure 4 now became main **new Figure 4**.

- Although the authors write it the title “... restores vision...”, in the manuscript they use in general “rescuing” vision. However, what is gone (rats are blind at 10 months) cannot be “rescued”, rather “restored”

We replaced “rescued” with “restored”, as suggested.

- The authors used male and female rats for their experiments which increases the meaning of their experiments. However, is there anything known if there are differences in the pathology of male and female RCS rats?

The MER-proto-oncogene tyrosine kinase (*Mertk*), bearing a spontaneous *loss-of-function* mutation in the RCS strains, is autosomic (chromosome 3) and no sex/gender differences have been reported.

- „.....restoring visual activities in fully degenerate retinas in the absence of residual photoreceptors and in the presence of an intense inner retina rewiring...”

Comment: photoreceptors are fully degenerated, retina is not fully degenerated

We corrected the sentence accordingly.

- “...aversion of nocturnal rodents to illuminated areas...” Comment: although rats have an aversion to illuminated areas, they are not nocturnal animals

Thank you for pointing it out. We removed “nocturnal”.

- Do the authors expect any effects if P3HT-NPs are injected intravitreally? This would be easier, especially for clinical applications

The primary targets of NPs were second-order neurons and not high-threshold RGC to allow for some extent of inner retina processing. Although an intravitreal administration is significantly easier and more reproducible than the subretinal injection, given the lack of radial diffusion across the retina thickness (Maya-Vetencourt *et al.*, *Nat Nano* 2020; this paper), it is very unlikely that our P3HT-NPs

(about 200 nm in diameter; see new Supplementary Figure 2) injected intravitreally can effectively cross the inner limiting membrane to reach RGCs.

Reviewer #3

This manuscript presents an investigation with conjugated polymer nanoparticles possessing intrinsic light-induced charge generation that allows for reinstating of physiological signals and visually driven activities. The work was performed in 10 months old RCS rats and this is very pertinent, as these old animals quite faithfully represent advance stages of retinal degeneration, with light-insensitive retinas and remodeled inner layers. This work is also a continuum of the previous work of the group (Maya-Vetencourt, J. F. *et al.* Nature Nanotechnology 15,698– 708, 2020) that evaluated the effects of the conjugated polymer nanoparticles P3HT NPs in 4- and 11-months old RCS rats. Here, the Authors found that P3HT-NPs distribute evenly in the subretinal space of young and old RCS rats, contact bipolar and horizontal cells after a single subretinal injection and lead to restoration of visually evoked potentials in the primary visual cortex and pupillary constriction light-driven behavioral responses (latency of escape from the illuminated area to darkness and percentage of time spent in the dark), and claimed that the extent of visual restoration positively correlated with the nanoparticle density and hybrid contacts with second-order retinal neurons.

Questions, concerns and suggestions:

1. There are no histological data to compare the status of the P3HT-particle injected animals and the glass-particle injected animals. Therefore, it is not possible to be sure that the surgeries went similarly in both cases. Quantification of the particle distribution is needed to ensure that the surgeries were identical for the same animals as those used in the physiological experiments.

The impact of surgery in P3HT and SiO₂ injected retinas were quite similar. We made a careful analysis of the effects of P3HT/SiO₂ NP injection surgery on photoreceptors/BPCs/HzCs mRNAs, inner retina rewiring and degree of retina inflammation (Supplementary Figures 5, 6 and 7 in the previous version). To extend and complete this assessment, we additionally studied the distribution of fluorescently labeled SiO₂-NPs using whole mount subretinally injected retinas analyzed by super-resolution confocal microscopy. The results of such an analysis, reported in the main **new Figure 3**, show that the distribution of P3HT-NPs and SiO₂-NPs are superimposable in terms of cluster size, retina coverage and nearest neighbor distance, agreement with previous reports (Maya-Vetencourt *et al.*, *Nat Nano* 2020). Moreover, in the new **Supplementary Figure 7**, we carried out conventional retina histology and in P3HT-NPs and SiO₂-NPs injected retinas, showing that the injection of the two kinds of NPs did not alter retinal structure versus age-matched untreated RCS retinas.

2. The surgery in RCS rats was found to clean some accumulation of cell debris resulting in improvement of visual function (Lorach *et al.*, *Sci Rep.* 2018; 8:11312). It is therefore important to evaluate how the surgery has removed all these debris and demonstrate that the observed effect is not due to this surgical effect. This question relates to point 1 and to the importance of verifying that the particle injections were identical with P3HT- and glass-particles.

We discussed in point 1 and demonstrated in Maya-Vetencourt *et al.*, *Nat Nano* (2020) that the P3HT- and SiO₂-NP injection surgeries had the same impact on retinal histology and inflammatory state (see the answer to point 1). The Lorach's paper does not apply to our study, as it was carried out in a different strain of RCS rats (pigmented and not pink-eyed), injected at a completely different age (1 month *versus* 10 months) and with a completely different surgical procedure (see the comparison of the surgical protocols below). The sham-injected group with inert NPs that fully mimic the distribution of P3HT-NPs is the gold standard control in pharmacology.

Nevertheless, even if we disagree with this criticism, we looked for debris in the subretinal space (as shown by Lorach *et al.*) and found that they are not different between untreated, and P3HT- or SiO₂-injected RCS rats, further supporting the statement that our surgery does not wash out the cellular debris in the subretinal space. We also want to emphasize that, in this paper, we are dealing with dystrophic retinas at the final stage, in which the cellular debris resulting from degeneration have been already cleaned up by microglial cells. Under these conditions, we guess the therapeutic washout proposed in the Lorach's paper would be ineffective. These data are now shown in the new **Supplementary Figure 7** and discussed in the Results (p. 8) and Discussion (p. 14) sections.

3. A major question concerns the differences between the efficient light intensity in vitro (>

10mW/mm²) and in vivo (5 lux). The authors should explain this discrepancy with some experimental data, otherwise, it supports the hypothesis explained in point 2 with an improved survival following the surgery. What is the rationale for the used light intensities: 5-50 lux for pupillary light reflex studies; 5 lux for light-dark box test. Please provide robust evidence about the efficacy at 5-lux conditions on the activation of the nanoparticles.

The discrepancy the Reviewer is mentioning between *in vitro* and *in vivo* studies refers to another work published previously (Maya-Vetencourt *et al.*, *Nat Nano*, 2020) and not to this paper. Nevertheless, we note that the issue is exhaustively explained in the very recent Benfenati & Lanzani "Reply to: Questions about the role of P3HT nanoparticles in retinal stimulation" *Nat Nano* (2021). The present paper is exclusively carried out *in vivo* with physiological light intensities. Regarding the pupillary reflex, the rationale for using 5 to 50 lux was to explore the response under mesopic/photopic conditions. Using a full range of luminances, one can explore how much of the activation of ipRGCs is due to the photoreceptor input or to the direct light sensitivity of melanopsin. The light-dark box was conducted at the lowest luminance which effectively elicited an escape behavior in healthy rdy rats of the same age.

4. In the previous paper in Nature Nanotechnology, the key element appeared to be the infrared sensitivity. Why this key factor was not used here?

This comment does not apply to this paper. The object of this work is to demonstrate that P3HT-NPs are capable to rescue vision even in animals with fully degenerated retinas, where no photoreceptors are surviving at all. The experiment the Reviewer refers to (Maya-Vetencourt *et al.*, *Nat Nano*, 2020) was aimed at demonstrating a direct role of NP in visual information independent of possible residual photoreceptors that escaped degeneration. In those experiments, infrared irradiation was used as a negative control, as it was unable to excite both photoreceptors and P3HT-NPs. The Reviewer likely refers to the red irradiation, which elicited VEPs ONLY in animals that received P3HT-NPs. A full description of this experiment can be found in Benfenati & Lanzani "Reply to: Questions about the role of P3HT nanoparticles in retinal stimulation" *Nat Nano* (2021). The scope of this paper is to use totally photoreceptor-free retinas and assess visual recovery in the full visible bandwidth.

5. The remodeling is not so major at this age (10 months old RSC rats) and it is possible that these animals still have some light perception as indicated by the response of the control animals in the light dark test.

We fully disagree. The remodeling is major at this age (see Figure 1). We can also quote Lorach's paper which declares that, in pigmented RCS rats (in which the degeneration is slower than in pink-eyed RCS rats), degeneration of photoreceptors is complete at postnatal day 180. The RCS animals that we injected at 10 months of age and analyzed up to 15 months had no photoreceptors and no visual perception. This was the main objective of replicating the data of Maya-Vetencourt *et al.* *Nat Nano* (2020) paper in 10-15 months old RCS rats and is clearly shown in new Figs. 1, 6 and 8 of the present manuscript. Immunohistochemistry and qPCR demonstrate the total absence of photoreceptors and VEP responses to flash stimuli are totally absent. The latency response of the non-injected or sham-injected RCS rats in the light-dark test does not indicate some residual light perception, but only the time to access the dark compartment by random exploration. This is confirmed by the total lack of preference for the dark. Moreover, the new results obtained with light-cued classical conditioning (see new Figure 9, described in the Results p. 11-12) further emphasize the absence of light perception in non-injected or sham-injected RCS rats.

6. Throughout the manuscript, the Authors talk about "restoring visual activities". They should clearly say which of the visual functions exactly have been restored. From the results, it appears that no visual function is restored except light perception! Visual perception is related to pattern perception and no data can clearly demonstrate pattern perception in this paper.

The new data on spatial vision and pattern perception (see Answer to Reviewer #2) together with the light-cued classical conditioning prove a "restoration of visual activities".

7. The link to the WHO is not the most appropriate reference for visual impairment due to inherited retinal diseases.

Thanks for the suggestion. We removed the WHO link from the text.

8. on line 69, the Authors say that "Overall, this approach is rather cumbersome and has severe limitations in performance" talking about optogenetics and without providing a reference for this statement. References measuring the theoretical visual acuity with optogenetics should be cited.

The Reviewer may have noted that we quoted the first application of optogenetics for the stimulation of RGCs in a patient with RP in whom the visual restoration was quite limited. In general, the "severe limitations" of optogenetics regard the safety and efficiency of gene therapy, as well as the potential risk of expressing very distant and immunogenic proteins on the membrane surface of neurons. We now provide references on this topic, as required. Regarding the approach being cumbersome, we wrote in the text that "*the inherent low sensitivity of the heterologous opsins requires an external camera coupled to light intensification goggles*", abolishing eye tracking movements and micro-saccades. Moreover, the rather limited area of expression of the opsin (often restricted to perifoveal RGSs in primates; see Chaffiol *et al.*, 2017 and 2022) may limit the quality and the extent of visual restoration and tunnel the visual field (Sahel *et al.*, 2021). In spite of these drawbacks, we think the the results obtained by the application of optogenetics to RP are quite remarkable.

9. on lines 165-166 "P3HT-NPs remained confined to the outer retina, replacing the lost photoreceptors, with no tendency to radially diffuse to the inner retina layers": please explain how/why this distribution is so selective? is it possible to generate pattern perception with so few particles?

The distribution of P3HT-NPs has no cell specificity, as demonstrated by the superimposable distribution of SiO₂-NPs. It is attributable to the following factors: (i) the cell barrier offered by second order retinal neurons to radial diffusion towards the inner retina; (ii) trapping of NPs in the subretinal space during spontaneous retina reattachment after surgery; (iii) no NP clearance due to the absence of blood vessels in the external retina. Regarding the possibility of generating pattern perception, see the answer to point 6 and Answer to Reviewer #2. Regarding the possibility of generating pattern perception with so "*few particles*", in the analysis of the retinal distribution of NPs (see point 1), we found a mean nearest neighbor distance of about 5 μm that is consistent with the recovery of spatial discrimination.

10. lines 219-223 "Photoreceptor markers were undetectable in all aged RCS rats, while the inner retinal markers were not affected by the microinjection of either P3HT-NPs or SiO₂-NPs". The Authors insist on the fact that the photoreceptors were no more there, so this statement sounds bizarre, please discuss this, possibly in the light of the photoreceptor survival demonstrated by Lorach *et al.* following a subretinal surgery because it washes out the cell debris below the retina.

The statement of lines 219-223 of the previous version was correct and based on the experimental evidence. The comparison with the Lorach's paper that does not apply to our study (see answer to point 2). Lorach *et al.* demonstrated a partial and transient preservation of few photoreceptors and amelioration of ERG up to 6 months after an energetic subretinal lavage performed in very young, pigmented RCS rats experiencing ongoing degeneration (treated at postnatal day 38). We used in 12/15 months-old pink eyed RCS rats for this very purpose. Moreover, the same Lorach's paper precisely states that degeneration of photoreceptors (that is slower in pigmented than pink-eyed RCS rats), is complete at postnatal day 180 (page 1, next to last line, of the Lorach's paper), fully consistent with our data and ruling out completely the possibility of the presence of surviving photoreceptors at the age in which we treated our animals.

Apart from these substantial differences (age and strain), the Lorach's procedure of subretinal washing is completely different:

"The subretinal surgery consisted of a transcleral incision and retinotomy of 1 mm, allowing for insertion of a 30G blunt cannula in the subretinal space. By gentle scraping and saline injection, the thick debris layer was removed in about 2 × 2 mm area, which brought the outer nuclear layer down to the RPE layer. No subretinal bleeding was observed during surgery. The relatively large sclerotomy allowed fluid and debris extrusion, as opposed to a small single puncture typically used for retinal detachment. "

from ours (compare underlined words):

"The subretinal injection of NPs was carried out as follows. A limbus-parallel conjunctiva dissection was performed with scissors for about 2 clock hours in the superior-temporal quadrant. A small incision (about 0.5 mm) through the sclera and

the choroid was carried out 1 mm from the *limbus*. Then, the retina was gently separated from the retinal pigment epithelium close to the incision using either a small amount of viscoelastic material or the tip of surgical scissors. SiO₂- or P3HT-NPs (1-2 μl; 1 mg/ml) were then injected through a 38G needle paying attention to penetrate the subretinal space tangential to the choroid and therefore efficiently detach the retina. The scleral incision was subsequently coagulated with diathermy and the conjunctiva repositioned over the scleral wound."

We always performed a small sclerotomy and a small single puncture, with no scraping and no fluid and debris extrusion. Last, but not least, we always introduced a proper sham-procedure group (SiO₂-NPs).

11. at different places throughout the manuscript the Authors talk about "visual rescue". First, the term "vision restoration" could better fit the effect they intend to describe; second, there is no demonstration of vision restoration but improved light perception. No temporal and spatial resolution.

See answer to point 6 and Answers to Reviewer #2.

12. lines 332-333 "Light stimuli, through P3HT-NPs, activated a correlated spiking activity in the optic nerve subserving a variety of functions": There is no measurement of spiking activity but only of visually evoked potentials. How can the Authors assert a variety of functions apart from light perception? There is no definition of spatial resolution neither temporal resolution.

The Reviewer is right. We modified the sentence. Regarding the rest of the comment, see the answer to point 6 and Answers to Reviewer #2.

13. in the discussion, when talking about "affecting the polarization" and commenting optogenetics, the Authors should also mention the mediation of depolarization with channelrhodopsins or hyperpolarization with halorhodopsins

This is exactly what we were referring to when writing "affecting the polarization" of the neuronal membrane. We changed the sentence to make it more understandable.

14. a big part of the discussion focuses on the previous data, ref 11, instead of the data from this study

We thank the Reviewer for this comment modified the Discussion accordingly. We have deleted the (single) paragraph referring to the previous study and paid more attention to the data of the present paper. If we can make a respectful comment, it seems to us that also the comments of this Reviewer are more concentrated on the 2020 paper than on the data from this study.

REVIEWERS' COMMENTS

Reviewer #1 (Remarks to the Author):

My comments were satisfactorily addressed and the manuscript improved significantly. I have no further comments.

Reviewer #2 (Remarks to the Author):

The authors addressed all minor issues and solved them to the full satisfaction of the reviewer. Moreover, the general questions were not only answered in the discussion (as suggested), but new in vivo experiments were performed with very convincing results were obtained, confirming the results presented in the first version of the manuscript and even going beyond.

I have no further objections and recommend this manuscript for publication.

Reviewer #3 (Remarks to the Author):

The authors addressed the majority of concerns and the manuscript improved significantly. Although not all concerns and suggestions have been considered, the modifications implemented in the revised version clarified some of the issues related to the results of this study and their scientific

ANSWERS TO REVIEWERS
Manuscript NCOMMS-21-40886A

Reviewer #1:

My comments were satisfactorily addressed and the manuscript improved significantly. I have no further comments.

Reviewer #2:

The authors addressed all minor issues and solved them to the full satisfaction of the reviewer. Moreover, the general questions were not only answered in the discussion (as suggested), but new in vivo experiments were performed with very convincing results were obtained, confirming the results presented in the first version of the manuscript and even going beyond. I have no further objections and recommend this manuscript for publication.

We thank both Reviewers #1 and #2 for their evaluation and constructive suggestions that gave us the opportunity to greatly increase the quality of the paper.

Reviewer #3:

The authors addressed the majority of concerns and the manuscript improved significantly. Although not all concerns and suggestions have been considered, the modifications implemented in the revised version clarified some of the issues related to the results of this study and their scientific interpretation.

We thank the Reviewer for her/his positive evaluation of the revised version of the paper. We would like to point out that, in the revision, we answered to virtually all questions and criticisms regarding the experiments present in the manuscript under evaluation, while we only discussed the issues regarding previous work.